# Modeling atmosphere-land interactions at a rainforest site - a case study using Amazon Tall Tower Observatory (ATTO) measurements and reanalyis data

Amelie U. Schmitt[1,2], Felix Ament[1], Alessandro C. de Araújo[3,4], Marta Sá[4], and Paulo Teixeira[4]

[1]Meteorological Institute, Center for Earth System Research and Sustainability (CEN), Universität Hamburg, Hamburg, Germany
[2]Climate Service Center Germany (GERICS), Helmholtz-Zentrum Hereon, Hamburg, Germany
[3]Empresa Brasileira de Pesquisa Agropecuária (EMBRAPA), Belém, Brazil
[4]Large Scale Biosphere-Atmosphere Experiment in Amazonia (LBA), Instituto Nacional de Pesquisas da Amazônia (INPA), Manaus, Brazil

**Correspondence:** Amelie Schmitt (amelie.schmitt@hereon.de)

**Abstract.** Modeling the interactions between atmosphere and soil at a forest site remains a challenging task. Using tower measurements from the Amazon Tall Tower Observatory (ATTO) in the rainforest, we evaluated the performance of the land surface model JSBACH focusing especially on processes influenced by the forest canopy.

As a first step, we analyzed whether high-resolution global reanalysis data sets are suitable to be used as land surface model forcing. Namely, we used data from the fifth generation ECMWF atmospheric reanalysis of the global climate (ERA5) and the Modern-Era Retrospective analysis for Research and Applications, Version 2 (MERRA-2). Comparing five years of ATTO measurements to near-surface reanalysis data, we found a substantial underestimation of wind speeds by about $1\,\mathrm{m\,s^{-1}}$. ERA5 captures monthly mean temperatures quite well but overestimates annual mean precipitation by 30 %. Contrarily, MERRA-2 overestimates monthly mean temperatures in the dry season (August - October) by more than $1\,\mathrm{K}$, while mean precipitation biases are small.

To test how much the choice of reanalysis data set and the reanalysis biases affect the results of the land surface model we performed spin-up and model runs using either ERA5 or MERRA-2 and with and without a bias correction for precipitation and wind speed and compared the results. The choice of reanalysis data set results in large differences of up to $1.3\,\mathrm{K}$ for soil temperatures and 20 % for soil water content, which are non-negligible especially in the first weeks after spin-up. Correcting wind speed and precipitation biases also notably changes the land surface model results - especially in the dry season.

Based on these results, we constructed an optimized forcing data set using bias-corrected ERA5 data for the spin-up period and ATTO measurements for a model run of two years and comparing the results to observations to identify model shortcomings. Generally, the shape of the soil water profile is not reproduced correctly, which might be related to a lack of vertical variability of soil properties or of the root density. The model also shows a positive soil temperature bias and overestimates the penetration depth of the diurnal cycle. To tackle this issue, potential improvements can be made by improving the processes related to storage and vertical transport of energy. For instance, incorporating a distinct canopy layer into the model could be a viable solution.

# 1 Introduction

The presence of vegetation - especially in forest canopies with tall trees - alters the exchange processes between land surface
and atmosphere. Forests influence the shape of the wind profile (Yi, 2008; de Souza et al., 2016; Santana et al., 2017) and the
structure of turbulence within and above the canopy layer (Chor et al., 2017; Dias-Júnior et al., 2019; Zahn et al., 2016). In tall
forests, the air close to the ground can even become decoupled from the air above the forest (Santana et al., 2018). Additional
heat storage in trees and the air within the canopy changes the surface energy balance (e.g. Oliphant et al., 2004; Lindroth et al.,
2010) and the transmission of radiation is affected by the leaf area and the canopy structure (e.g. Hardy et al., 2004). Forests
also have an impact on global carbon and water cycles. Evapotranspiration is particularly enhanced in rainforests (de Oliveira
et al., 2018; Costa et al., 2010), which leads to a moistening of the atmosphere and increased cloud cover (Wright et al., 2017).

These processes at the canopy scale are very complex and thus difficult to represent in coupled atmosphere-land models. A
few land surface models (LSMs) - such as CLM (Bonan et al., 2018) or ORCHIDEE (Chen et al., 2016), for example - have
implemented parameterizations of vertically resolved processes in forest canopies in the past years. However, many LSMs
still incorporate rather simple parameterizations of canopy effects. The aim of this study is to evaluate the performance of an
LSM without a resolved canopy layer in a region with a tall forest canopy. For this purpose, we choose the land surface model
JSBACH and compare model results to measurements from the Amazon Tall Tower Observatory (ATTO) located at a rainforest
site.

Observations from tropical forests have been previously used to evaluate specific aspects of LSMs, for example to test
different parameterizations of rainfall intercepted by forest canopies (Wang et al., 2007; Fan et al., 2019). Anwar et al. (2022)
used measurements from different FLUXNET sites in the Amazon rainforest to test the performance of two land-surface
hydrology schemes in the CLM model. They found that both schemes were able to reproduce the shape of the seasonal cycle
of turbulent fluxes but failed in capturing their correct magnitude. Other studies used measurements from canopies in the
mid-latitudes - which are dominated by deciduous broadleaf and evergreen needleleaf forests - for example to evaluate the
performance of multi-layer canopy representations in LSMs (Ma and Liu, 2019; Bonan et al., 2021). Studies using specifically
the JSBACH model have focused on calibrating stomatal conductance using FLUXNET data from evergreen needleleaf forests
(Mäkelä et al., 2019) or on the global evaluation of a canopy heat storage parametrisation (Heidkamp et al., 2018). In this study,
we use a site-level setup of the JSBACH LSM to evaluate the model performance at a rainforest site with a special focus on
canopy processes.

LSMs are not only used in coupled model setups but can also be run offline using external data as atmospheric forcing.
Feedbacks between land and atmosphere in coupled models make it more difficult to asses single processes and thus it is
beneficial to use offline simulations to isolate processes when evaluating LSMs (e.g. Decharme et al., 2019). In addition,
uncertainties of the LSM originating from errors in the atmospheric model can be avoided by carefully choosing a forcing
data set. Besides observations, reanalysis data are frequently used for offline and site-level simulations, which aim at testing
the model's performance with respect to different parameterizations (Brun et al., 2013; Knauer et al., 2017) or initialisation
data sets (Ardilouze et al., 2017), for example. To minimize the impact of forcing data on the model results, some studies

also use bias corrected reanalysis data sets, such as the WFDEI meteorological forcing data set (Weedon et al., 2014). When evaluating the global performance of the ISBA land model, Decharme et al. (2019) used atmospheric forcings based on two different reanalysis data sets. The results showed large differences with respect to hydrological variables, which underlines the importance of the choice of forcing data sets for the evaluation of land surface models.

After initialization, an LSM undergoes an adjustment process until an equilibrium between external forcing and the simulated land surface fluxes is reached. The length of this so-called spin-up period can reach several years (Yang et al., 1995) but requires shorter time spans for locations with large precipitation amounts (Lim et al., 2012; Yang et al., 2011). Here, we use a spin-up period of ten years following the example of Heidkamp et al. (2018) who evaluated land surface fluxes in the LSM JSBACH. For our JSBACH site-level simulations we also intend to partly use reanalysis data since ATTO measurements contain many data gaps. We use two different high-resolution global reanalyses (ERA5 and MERRA-2) to compare and minimize the impact of the forcing data set on the model results. As a first step, we compare near-surface meteorological data reanalyses to ATTO measurements to answer the following questions: How well do the reanalyses reproduce meteorological conditions - ranging from hourly to yearly scales - at the ATTO site? And are the forcing data thus generally suitable as forcing data sets for the land surface model? How does the choice of forcing data set affect the results of the land surface model? For this purpose, we compare model results using the two different reanalyses as forcing for the model spin-up run. Also, we test how a correction of reanalysis biases changes the model results. Based on the findings from these preparative analyses we perform a model run using an optimized forcing data set (consisting of air temperature and humidity, wind speed, precipitation and incoming long- and shortwave radiation) and compare the results to ATTO measurements focusing on the following questions: How well does JSBACH reproduce the temporal evolution of the model output variables soil water content, soil temperatures and turbulent heat fluxes? Can we identify specific model shortcomings, which could be improved in future model versions?

Observations from the ATTO site, data from the ERA5 and MERRA-2 reanalyses and the JSBACH land surface model are described in Sect. 2. Results of the comparison between reanalysis data and ATTO measurements are presented and discussed in Sect. 3.1. Section 3.2 contains results and a discussion of JSBACH model runs, which are used for sensitivity studies (Sect. 3.2.1 and 3.2.2) and to identify model shortcomings (Sect. 3.2.3), followed by a summary and conclusions in Sect. 4.

## 2   Data and model description

### 2.1   Amazon Tall Tower Observatory

#### 2.1.1   Measurements at the ATTO site

The Amazon Tall Tower Observatory (ATTO) is a scientific research facility in Brazil with a focus on interactions between the rainforest and the atmosphere. An extensive description of the characteristics of the ATTO site can be found in Andreae et al. (2015). Here, we summarize the most important aspects. The ATTO site is located in the central Amazon at an altitude of 130 m, roughly 150 km north-east of Manaus (Fig. 1). The area is covered by terra firme forest with an average canopy

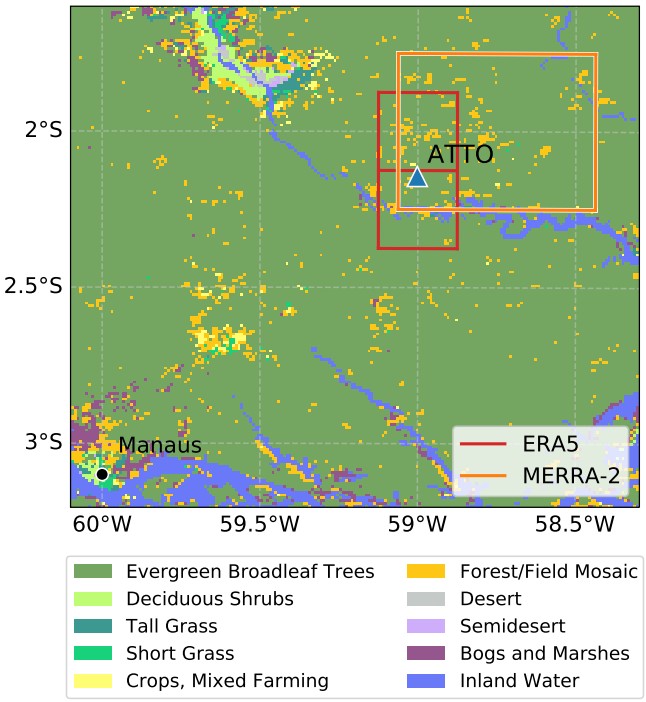

**Figure 1.** Location of the ATTO site (blue triangle) and reanalysis grid boxes used for comparison in this study. The background shows GLCC land cover types derived using the BATS scheme for 1992 data (Loveland et al., 2000).

height of about $37.5\,\mathrm{m}$. In this study, we use measurements from the $81\,\mathrm{m}$ high walk-up tower, which is located at $2.144°$ S and $59.002°$ W.

We use five years of ATTO measurements from 2014 to 2018. For an overview of the measurements used in this study see Table A1. Incoming shortwave and longwave radiation, precipitation, pressure, as well as turbulent sensible and latent heat fluxes are measured at the top part of the tower. Air temperature, relative humidity and wind speed are measured at several heights above and within the forest canopy. Since the reanalyses used for comparison in Sect. 3.1 do not have a separate resolved canopy layer, we use only measurements above the canopy top ($\geq 36\,\mathrm{m}$) for the comparison. Additionally, air

temperatures within the canopy at $1.5\,\mathrm{m}$ above the ground are used to calculate temperature differences between the soil and atmosphere (see Sect. 3.2.3). Measurements of soil moisture and temperature are conducted close to the walk-up tower and are available at several depths (see Table A1).

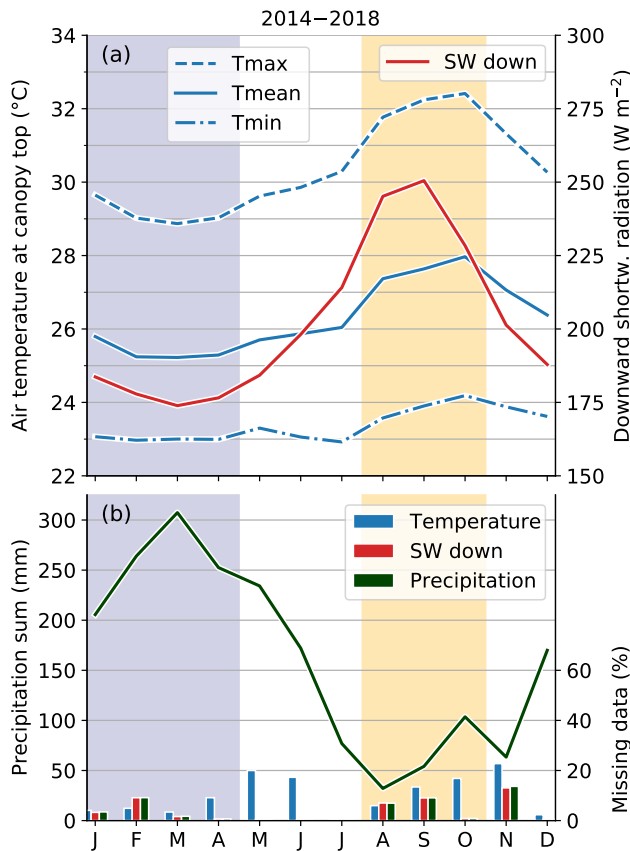

**Figure 2.** Average annual cycles based on ATTO measurements of the years 2014 to 2018. (a) Mean temperature and average daily maximum and minimum temperatures measured at 36 m height (blue) and incoming shortwave radiation (red). (b) Precipitation (dark green) and fraction of missing data for all parameters displayed in this figure (bars). Shaded areas denote the wet (blue) and dry (yellow) seasons.

### 2.1.2 Data preparation

For comparison with reanalysis data and for model forcing ATTO measurements are aggregated to hourly values. We apply a
100 linear interpolation to fill small data gaps of up to 10 minutes for temperature, humidity, wind, pressure and radiation and up to 1 minute for precipitation. Larger data gaps were caused by interruptions of the power supply or by general sensors issues. To account for these gaps we mask out times when any of the considered ATTO data are missing in the reanalysis data as well (Sect. 3.1). Wind measurements at 43.1 m height are only available for the years 2014 and 2015 and were therefore not used for the general statistical analysis. We convert the measured relative humidity to specific humidity and the air pressure measured
at 81 m height to surface pressure. More details are presented in Appendix B.

### 2.1.3 Seasonality

Being located at around 2° S, the climate at ATTO is tropical humid and strongly influenced by the location of the Inter-Tropical Convergence Zone, leading to pronounced wet and dry seasons. While Andreae et al. (2015) strictly divide the year into a wet and a dry season, Saturno et al. (2018) also consider transition zones between the seasons.

In this study we calculate seasonal means and thus aim for relatively homogeneous meteorological conditions within a season. Average annual cycles of air temperature, incoming shortwave radiation and precipitation are shown in Fig. 2. For our purposes, we define the wet season as the months from January to April, which are characterized by large precipitation sums exceeding 200 mm per month as well as monthly mean temperatures and shortwave radiation below the annual means. Accordingly, the dry season is defined as the months from August to October with precipitation sums below 100 mm per month. Air temperatures in the dry season are on average more than 2 K higher than in the wet season. The wind speed shows no distinct annual cycle and seasonal changes are small. Due to the latitude there are only small variations of less than 15 minutes of the day length, which facilitates the interpretation and comparison of average diurnal cycles.

## 2.2 Reanalysis data

We utilize only those global reanalysis data sets that provide data at least hourly, which enables us to analyze diurnal cycles. Specifically, the selected data sets include ERA5 (Sect. 2.2.1) and MERRA-2 (Sect. 2.2.2), which also have a relatively high spatial resolution of less than 70 km.

### 2.2.1 ERA5

ERA5 is the fifth generation reanalysis produced by ECMWF based on version Cy41r2 of the IFS model (Hersbach et al., 2020). Model output is available every hour on a 0.25°x 0.25°grid with 137 vertical levels. Mean total precipitation rate, incoming longwave and shortwave radiation, sensible and latent heat flux from the ERA5 surface data set (Hersbach et al., 2018) and air temperature, specific humidity, horizontal wind components, and pressure from ERA5 on model levels (Hersbach et al., 2017) for the years 2007 to 2018 were downloaded from the Copernicus Climate Change Service (C3S) Climate Data Store. Since ATTO is located close to the border of two intersecting ERA5 grid boxes (see Fig. 1), we average the values of those grid boxes using inverse distance squared weighting.

The IFS model calculates exchange processes at the land surface using the Tiled ECMWF Scheme for Surface Exchanges over Land with revised Hydrology component (HTESSEL; e.g. van den Hurk et al., 2000; Balsamo et al., 2009). Each grid box consists of separate tiles - e.g. for water, bare ground, high and low vegetation (ECMWF, 2016b). The tile fractions are based on the USGS Global Land Cover Characteristics (GLCC) data set with a 1-km resolution, which is derived from one year of remote sensing data from the Advanced Very High Resolution Radiometer (AVHRR) in 1992 (Loveland et al., 2000). The GLCC data are shown as background in Fig. 1. The grid boxes used for comparison with ATTO measurements are mostly covered by evergreen broadleaf trees with fractions of 97.5 % and 99.6 % for the northern and southern grid box, respectively.

We calculate the geometrical height of the model levels using the surface pressure, as well as temperature and specific humidity on model levels according to Eq. 2.22 of the IFS model documentation (ECMWF, 2016a). For the study period and location the heights of the lowest model levels are approximately 10, 32 and 57 m. Some of the ERA5 variables are provided as instantaneous values. For comparison with ATTO and MERRA-2, we average two successive values to obtain an estimate of the hourly mean.

### 2.2.2 MERRA-2

MERRA-2 is the second version of the Modern-Era Retrospective analysis for Research and Applications (Gelaro et al., 2017) produced using the GEOS-5 atmospheric general circulation model by GMAO (NASA's Global Modeling and Assimilation Office). Model output is available every hour on a 0.5°-latitude and 0.625°-longitude grid with 72 vertical levels. The following variables from MERRA-2 for the years 2007 to 2018 were obtained from Global Modeling and Assimilation Office (GMAO) (2015): Incoming longwave and shortwave radiation; sensible and latent heat flux; pressure; air temperature, specific humidity and horizontal wind components at 10 and 50 m height; bias corrected precipitation.

The location of the MERRA-2 model grid box used for comparison with ATTO measurements is displayed in Fig. 1. As for ERA5, MERRA-2 land cover is based on the GLCC data set (Loveland et al., 2000). We calculate the pressure at model levels using the pressure thickness variable and apply the same method as for ERA5 to calculate the geometrical height of the model levels. For the study period and location the height of the lowest model levels is approximately 68 m.

To improve the soil moisture calculations in MERRA-2 the model-generated precipitation is corrected using observations (Reichle et al., 2017). In the ATTO region, gauge-based precipitation estimates from the CPCU data set (Unified Gauge-Based Analysis of Global Daily Precipitation) are used to replace the model precipitation. However, the number of rain gauges in the Amazon region has decreased in the last decades, resulting in an uncertainty increase of the CPCU data set (Reichle et al., 2017).

### 2.3 Modeling setup

### 2.3.1 The land surface model JSBACH

We use the land surface model JSBACH (Giorgetta et al., 2013) to simulate fluxes of energy, water and momentum between the land surface and the atmosphere. We perform our model simulations using JSBACH4, which is a more flexible re-implementation of JSBACH3 and can be used within the global models MPI-ESM1.2 (Mauritsen et al., 2019) or ICON-A (Giorgetta et al., 2018; Jungclaus et al., 2022), as well as a stand-alone model forced by external data (e.g. Nabel et al., 2020).

JSBACH uses a fractional approach to represent vegetation classes. Each grid cell is divided into tiles with 11 different vegetation classes. The grid box used for the model simulations at the ATTO location contains 5.6 % lake and river areas an the remaining land fraction is covered by 99.3 % tropical evergreen trees with a root depth of 1.95 m. The soil depth at this grid box is 2.23 m, covering the upper 4 of 5 model layers. These characteristics are based on the data set of land surface parameters

derived by Hagemann (2002) on a T63 spectral grid with 192x98 (lon,lat) grid points, which corresponds to a grid cell size of about 200 km at the considered latitude.

Vertical soil water transport within the model is calculated using the one-dimensional Richards equation (see e.g. Ekici et al., 2014, , Eq. 2), which incorporates vertical diffusion, percolation from gravitational drainage and sources and sinks. The only depth-dependent sink term is transpiration, which depends on the root density. The surface energy balance in JSBACH is closed within the uppermost soil layer - meaning that the soil temperature between 0 and 6.5 cm is considered as the surface temperature.

For comparison with point measurements at ATTO, we use a site-level setup of JSBACH consisting of a single grid box. The required meteorological variables for model forcing are air temperature, specific or relative humidity, precipitation, wind speed, as well as incoming longwave and shortwave radiation. We use ATTO measurements, as well as reanalysis data from ERA5 and MERRA-2 as forcing data sets to obtain the results in Sect. 3.2. The forcing is applied at each model time step of 1 hour.

**2.3.2 Preparation of model forcing data**

We perform two separate 10-year model spin-up runs using ERA5 data from the lowest model level of about 10 m height and MERRA-2 data from 10 m height of the years 2007 to 2016. After the spin-up, we perform model runs for the years 2017 and 2018 using forcing data from either ERA5, MERRA-2, or ATTO measurements from 4 m (temperature and specific humidity), 13 m (wind), and roughly 40 m (radiation and precipitation) above the forest top.

ATTO measurements contain data gaps that need to be filled before the data can be used as model forcing. We apply the following methods: 1) Small gaps of up to 4 hours (2 hours for precipitation and shortwave radiation) are filled by linear interpolation. 2) Medium sized gaps of up to 6 days are filled with data from the day before and after the gap. Only days with at least 12 hours of valid measurements are used for filling. 3) Large gaps are filled with a monthly mean diurnal cycle, which we calculate by averaging only days without missing values for each calendar month of the years 2017 and 2018. Time periods 190 during which missing data have been interpolated are masked out in the interpretation of the model results in Sect. 3.2.

## 3 Results and discussion

### 3.1 Comparison of reanalysis data to ATTO measurements

To evaluate whether reanalysis data are suitable forcing data sets for a land surface model we compare ATTO measurements closely above the forest top to reanalysis data at the corresponding heights. For this purpose, ATTO measurement heights are 195 specified as heights above forest (a.f.) relative to the forest top of about 37.5 m height. Here, we focus on the forcing data required for JSBACH, which are air temperature, specific humidity, wind speed, precipitation, and incoming shortwave and longwave radiation. As described in Sect. 2.1.2 hours with measurement gaps are also masked out in the reanalysis data sets.

As a first step, we consider annual mean temperatures and precipitation sums, which are widely used to characterize local climates. ERA5 annual mean temperatures at 10 m height between 2014 and 2018 agree almost perfectly with the ATTO values measured at 18 m above the forest - both with mean values of 26.1 °C and an RMSD of 1.4 K. Compared with ATTO measurements, MERRA-2 is generally too warm with annual average temperatures of 26.9 °C and a larger RMSD of 2.0 K. For annual precipitation sums we see the opposite, with very good agreement between ATTO (1560 mm) and MERRA-2 (1540 mm) and a strong overestimation of ERA5 with 2030 mm per year. A more detailed comparison of average annual and diurnal cycles is presented in Sect. 3.1.1 for wind, in Sect. 3.1.2 for precipitation and radiation and in Sect. 3.1.3 for temperature and humidity.

### 3.1.1 Wind

The most apparent feature of the wind speed annual cycle is the large underestimation by both reanalysis data sets (Fig. 3a). Averaged over the years 2014 to 2018, ERA5 shows a negative bias of about -0.8 $ms^{-1}$ and MERRA-2 an even larger one of -1.2 $ms^{-1}$. The bias does not change significantly during the course of the day (Fig 3b). The shape of the diurnal cycle is well reproduced by both reanalyses, while ERA5 does a better job a capturing the shape of the annual cycle. Average wind profiles indicate that the bias prevails also for larger heights up to 36 m above the forest (Fig 3c).

Our results indicate large biases at heights close to the forest canopy and could thus simply be related to the shape of the wind profile at these heights. To analyze whether the bias extends to larger heights, we examined measurements from the top of the ATTO tall tower at 285 m above the forest top, which is available from the ATTO data portal (attodata.org) for the time period from March 2018 to the end of 2019. At this higher altitude we also found negative biases in the order of -1 $ms^{-1}$ for both reanalyses (not shown), indicating that wind speed biases are a significant issue within the lower parts of the ABL in this region.

Using these reanalysis data, which underestimate the observed wind speed, to force a land surface model, we expect to see an impact on turbulent fluxes and thus also on related soil quantities. Since turbulent heat fluxes scale with wind speed, an underestimation of the latter would initially result in an decrease of sensible and latent heat fluxes, which then increases the surface temperature. However, a higher surface temperature increases sensible heat fluxes and thus the overall impact on surface and soil temperatures is difficult to estimate.

Wind speed biases with the same order of magnitude as our results have also been found in various previous studies. Jourdier (2020) compared ERA5 and MERRA-2 wind speeds to measurements between 55 m and 100 m height at eight locations in France. They found that ERA5 wind speeds were generally about 1 to 1.5 $ms^{-1}$ lower than those from MERRA-2. ERA5 wind speeds agreed better with measurements from the northern flat terrain, while those from MERRA-2 agreed better in southern mountainous regions. The dependence of the sign of the wind speed bias on the region has also been found in a study by Staffell and Pfenninger (2016). Their results for MERRA-2 indicate a line dividing Europe with negative wind speed biases in the Mediterranean region and positive biases around the North and Baltic Seas. Carvalho (2019) compared MERRA-2 to wind speed measurements all over the globe and also found that the sign of the 10-year mean bias over land varied for different regions. In the northern Amazon region, the found a general underestimation of about -1 $ms^{-1}$, which is in line with our

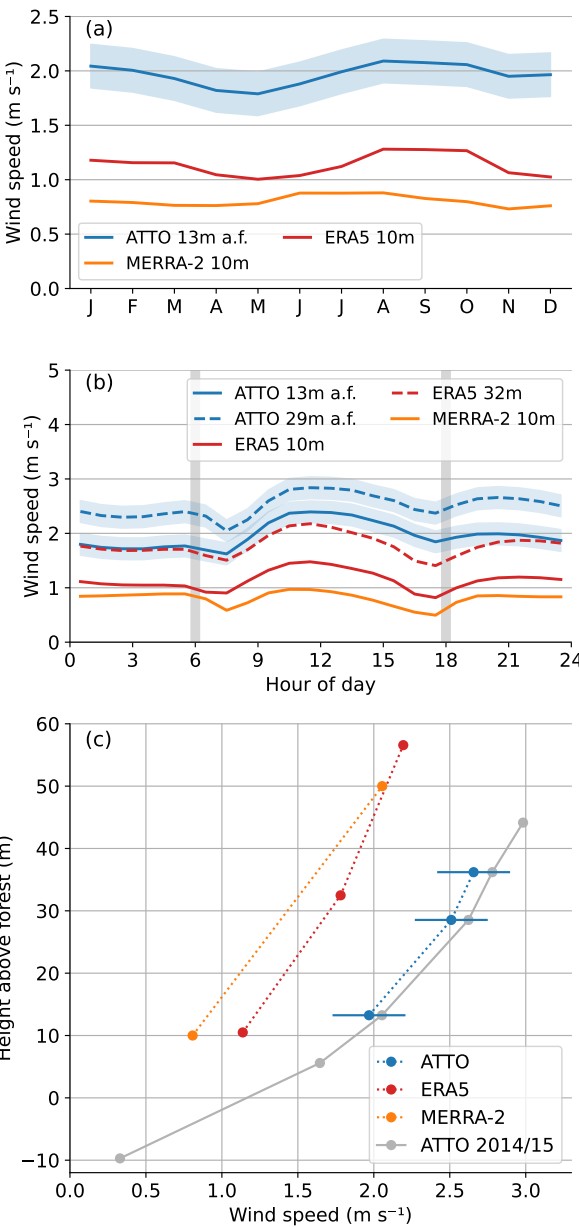

**Figure 3.** Average annual (a) and diurnal (b) cycles of wind speed averaged over the years 2014 to 2018 based on ERA5 (red) and MERRA-2 (yellow) reanalysis data and ATTO measurements (blue). Gray vertical lines mark the times of sunrise and sunset, respectively. Wind speed profiles based on the same data are shown in panel (c). The grey line shows a profile with additional heights, including wind measurements at 6 m above the forest, which are only available for the years 2014 and 2015. Measurement uncertainty in all panels is denoted as blue shading and errorbars, respectively.

results. Gualtieri (2021) compared mast measurements to ERA5 wind speeds and found large negative biases of up to -3 ms$^{-1}$ at mountain sites and a positive bias in a forest site. They concluded that reanalyses with a spatial resolution of several tens of kilometers have problems accurately reproducing wind speeds at sites with high variation of topography and land use.

### 3.1.2 Precipitation and radiation

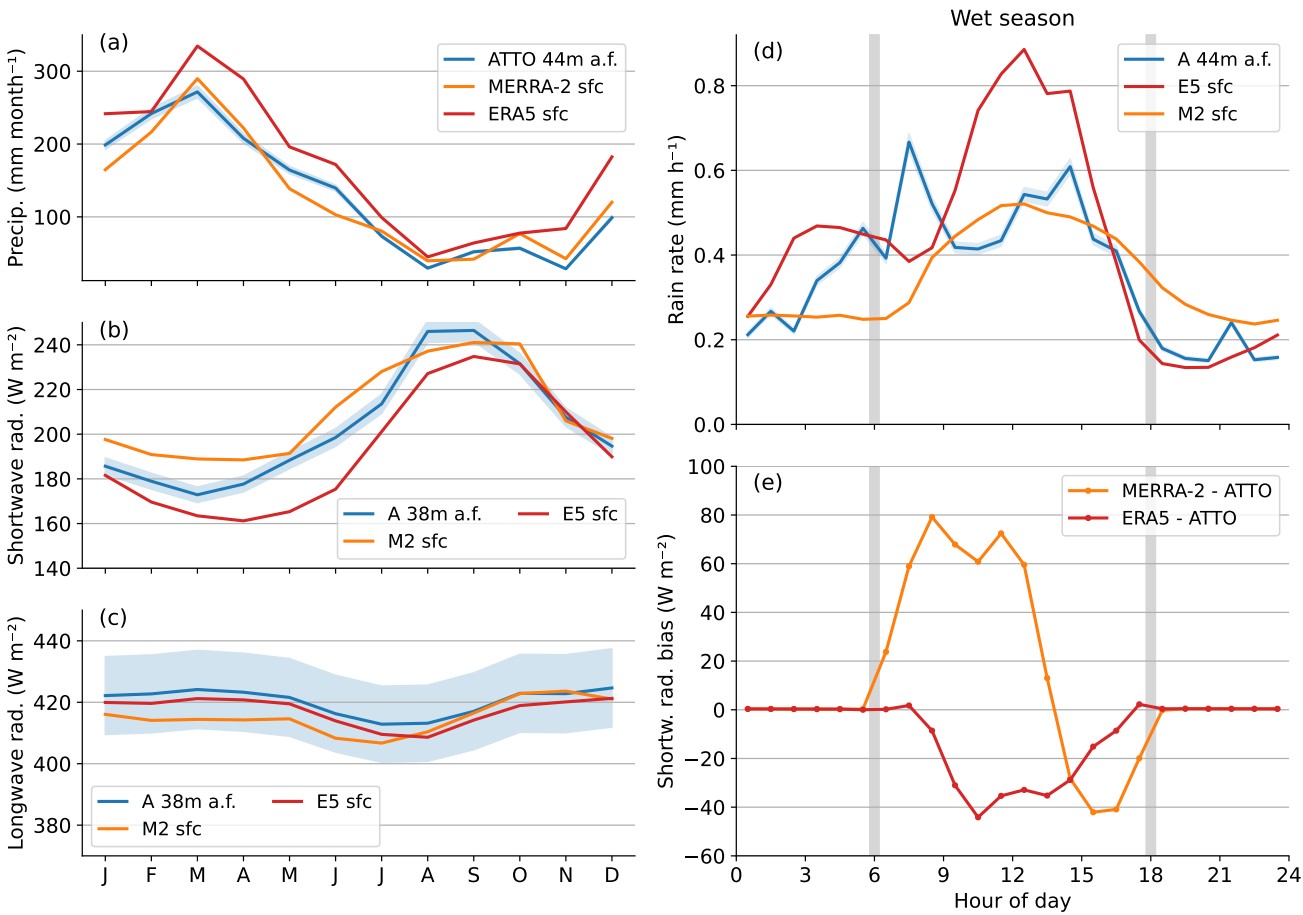

**Figure 4.** Average annual cycles of precipitation (a), downward shortwave (b) and longwave radiation (c) averaged over the years 2014 to 2018 based on ERA5 (red) and MERRA-2 (yellow) reanalysis data and ATTO measurements (blue). Average diurnal cycles for (d) rain rate and (e) incoming shortwave radiation bias. The curves represent hourly means from the wet season (JFMA) of the years 2014 to 2018. Gray vertical lines mark the times of sunrise and sunset, respectively. Shaded areas in all panels denote the ATTO measurement uncertainty.

Both ERA5 and MERRA-2 are able to reproduce the seasonality of precipitation and incoming shortwave radiation (Fig. 4a,b). The amplitude of the annual cycle of longwave radiation is small and biases for both reanalyses are within the ATTO measurement uncertainty (Fig. 4c). Precipitation sums from ERA5 are too large in all seasons with an overestimation of about 20 %

in the wet season and of 45 % in the dry season. MERRA-2 captures monthly precipitation sums quite well, which can be
attributed to the correction of precipitation biases over land within the model using a gauge-based data set (see Sect. 2.2.2).

The diurnal cycle of precipitation measured at ATTO in the wet season (Fig. 4d) shows two distinct maxima - one shortly after sunrise and one in the afternoon around the time of the temperature maximum. A similar shape can be found in the dry season but with much smaller amplitudes (not shown). MERRA-2 captures the amplitude of the afternoon peak quite well, while ERA5 overestimates the maximum rain rate by about 40 %. ERA5 also shows a negative shortwave radiation bias during
the day (Fig. 4e), which indicates an overestimation of the cloud cover. MERRA-2 fails to reproduce this early morning peak. ERA5 does produce a secondary rainfall maximum, however it occurs earlier in the night between 2:00 and 6:00 local time.

The afternoon peak in continental rainfall is related to the diurnal cycle of solar forcing with land surface heating causing an ABL growth and the formation of convective clouds (see e.g. Yang and Smith, 2006). The early morning precipitation peak can be observed more frequently over ocean regions. A common explanation is that nighttime cooling of the cloud-top causes a
thermal destabilisation of the upper cloud, which then increases convection and precipitation (see e.g Randall et al., 1991). To further analyze the occurrence of the early morning peak we evaluated rainfall data from the IMERG data set in a larger region around ATTO (for details see Appendix C) and compared the results to data from the two reanalyses. The IMERG results indicate that nighttime precipitation occurs more frequently in the region north-east of ATTO, while precipitation after sunrise between 6:00 and 8:00 is more patchy. The location of the morning maximum is also quite variable between the considered
years (not shown). For MERRA-2, the analysis of regional patterns of early morning precipitation reveals that a morning maximum can be found at grid points located about 100 km to the east. This gives a hint that MERRA-2 might just produce early morning precipitation at a slightly wrong location. Further evidence can be found in a study by Sato et al. (2009) who showed that the timing of local precipitation maxima strongly depends on the model resolution.

### 3.1.3   Temperature and humidity

Monthly mean annual cycles of near-surface air temperature based on ATTO measurements and reanalysis data are shown in Fig. 5a. The comparison reveals that ERA5 accurately captures the shape of the annual cycle with a negligible bias. However, MERRA-2 shows a good agreement only during the wet season, while model results in the dry season are too warm (+1.1 K). It is evident from the diurnal cycles in Fig. 5b that the positive temperature bias for MERRA-2 in the dry season is mainly driven by too high daily maximum temperatures. MERRA-2 maximum temperatures exceed the measurements by about 2.5 K,
while nighttime temperatures agree well with measurements. Also, MERRA-2 maximum temperatures in the wet season show a much better agreement with ATTO measurements (Fig. 5c), which is in line with the smaller differences of monthly means (Fig. 5a) in this season.

The shape of the annual cycle of specific humidity is reproduced well by ERA5, but with an underestimation of roughly $1 \, \mathrm{g \, kg^{-1}}$ in all months (Fig. 5d). Similar to temperature, MERRA-2 shows a good agreement of monthly mean specific humidity
with small biases only in the wet season. In the dry season, values are underestimated by about $-0.9 \, \mathrm{g \, kg^{-1}}$. We further examine the humidity biases by analysing diurnal cycles for the dry and wet seasons separately (Fig. 5e,f). It is most striking that the diurnal cycles show distinctly different shapes, with a maximum of the humidity measured at ATTO in the late morning in

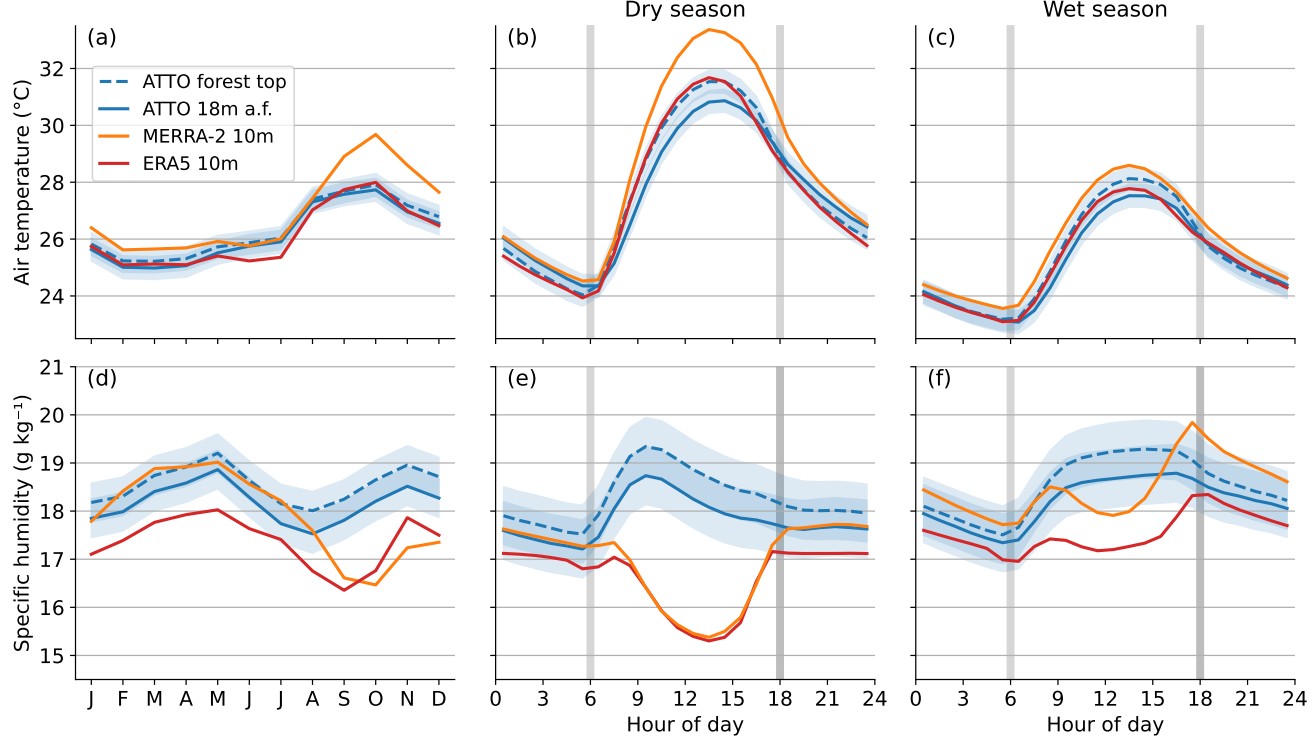

**Figure 5.** Average annual cycles of the years 2014 to 2018 for air temperature (a) and specific humidity (d) closely above the forest top and average diurnal cycles for temperatures (b,c) and humidity (e,f). The diurnal cycles are based on hourly means from the dry (b,e) and the wet (c,f) seasons. Shaded areas denote the ATTO measurement uncertainties. Gray vertical lines mark the times of sunrise and sunset, respectively.

the dry season and later in the afternoon in the wet season. In contrast, specific humidity values of both reanalyses start to decrease in the morning with a (local) minimum in the early afternoon. For ERA5, this general underestimation of daytime
specific humidity is the reason for the underestimation of monthly means in all months observed in Fig. 5d. For the same reason, MERRA-2 underestimates monthly mean humidity in the dry season. However, the diurnal cycle in the wet season indicates that MERRA-2 humidity is always about $0.9\,\mathrm{g\,kg^{-1}}$ larger than ERA5 humidity. Thus, the overestimation of nighttime humidity values compared to ATTO measurements compensates the underestimation in the afternoon, resulting in a negligible overall bias of monthly means in the wet season.
In the following, we discuss possible reasons for the differences between observed and modeled temperature and humidity. The overestimation of the daily maximum temperatures in the dry season by MERRA-2 could be related to an underestimation of cloud cover. However, incoming shortwave radiation from MERRA-2 does not show a considerable bias in the dry season (Fig. 4b). Another hypothesis is that the bias of daily maximum temperature is related to the vertical structure of the ABL, i.e. vertical mixing and stability. Testing this hypothesis, however, would require more investigations with measurements span-

ning the whole ABL column, e.g. from radiosonde measurements. Such measurements are unfortunately not a part of regular measurements conducted at the ATTO site.

The diurnal cycles of specific humidity showed a maximum during midday for ATTO measurements but a minimum for both reanalyses. The processes leading to such a humidity minimum during midday have been described in detail by Brümmer et al. (2012), for example. Evaporation starting in the morning leads to an increase of the absolute humidity. However, when

the atmospheric boundary layer (ABL) grows during the day, entrainment and downward-mixing of dryer air from above cause a decrease in humidity. These mixing processes lead to a humidity minimum that coincides with the time of the temperature maximum. This process is modeled by the reanalyses, but it appears that this is not what happens in reality at the ATTO site. The different observed shape of the diurnal cycle with a maximum at midday could have two possible reasons: 1) evapotranspiration is stronger than modeled by the two reanalyses, or 2) vertical mixing-processes within the ABL are weaker than in the

reanalyses. To test the first hypothesis we analysed the diurnal cycles of latent heat fluxes (not shown) and found that during the dry season, maximum measured latent heat fluxes were indeed larger than those modeled by the two reanalyses. However, during the wet season ERA5 underestimates the measured fluxes while MERRA-2 shows a strong overestimation of the latent heat fluxes. Since the two reanalyses do not agree on the sign of the flux bias, it is unlikely that the different shapes of the humidity diurnal cycles are mainly related to evapotranspiration.

The shape of the humidity diurnal cycles is also related to the ABL growth. It is possible that vertical mixing within the ABL or entrainment at the ABL top in the reanalyses are too strong and thus the ABL grows too quickly in the morning. Testing this hypothesis would require measurements spanning the whole boundary layer depth. In a study conducted by Dias-Júnior et al. (2022), data from a campaign at ATTO in November 2015 were used to compare ABL heights derived from ERA5 with those obtained from radiosonde and ceilometer measurements. Their findings indicate that the ERA5 ABL grows faster in the

morning and is larger than the measured values after 9:00 local time. The average maximum ABL height from ERA5 exceeds the measured values by more than 200 m. We checked MERRA-2 ABL heights for the corresponding periods and found an almost identical ABL growth rate in the morning with an even larger maximum exceeding ERA5 by about 200 m. These results are well in line with our second hypothesis and thus observed humidity biases are probably related to the growth rates of the ABL.

## 3.2 JSBACH model simulations

The comparison of near-surface atmospheric variables in the previous section revealed a mostly good agreement between reanalysis data and measurements at the ATTO site but also notable biases for certain variables. In this section, we examine the impact of these biases on JSBACH model results if reanalysis data are used for model spin-up (Sect. 3.2.1) or as forcing of the subsequent model run (Sect. 3.2.2). Based on the conclusions of these two sections we then set up a model run with optimized

forcing, which is based on bias-corrected ERA5 data and ATTO measurements (see Sect. 3.2.3 for details) . By comparing model results to ATTO measurements of soil variables and surface fluxes, we then aim to identify possible model shortcomings (Sect. 3.2.3).

### 3.2.1 Impact of the choice of spin-up data set

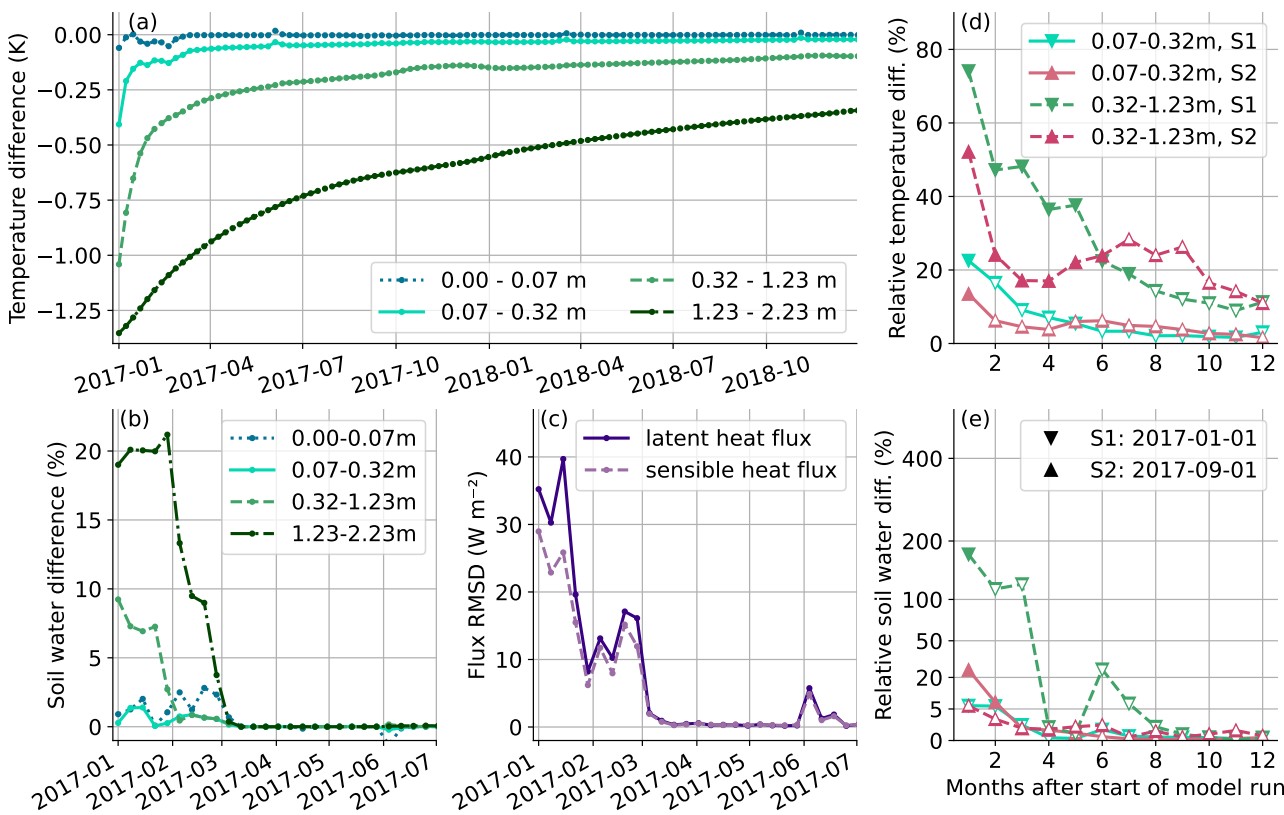

**Figure 6.** Consequences of different spin-up data sets on soil conditions: model results using ERA5 data for spin-up minus those using MERRA-2. Time series of differences for (a) soil temperature, (b) soil water content at different depths and (c) RMSD for turbulent fluxes. Corresponding differences between ERA5 and MERRA-2 spin-up relative to model biases (differences between model results and measured ATTO data, see also Sect. 3.2.3) are shown in panels (d) for soil temperature and (e) for soil water. Results are presented for two different starting times of the model run after spin-up in the wet (S1: January 2017) and dry (S2: September 2017) seasons. Empty symbols in (d) and (e) indicate that differences between ERA5 and MERRA-2 spin-up are below 0.2 K for soil temperature or below 0.7 % for soil water content (for S1 compare values in (a) and (b), respectively).

Land surface models require a spin-up period of several years to reach an equilibrium state for the soil water content. Since on-site measurements are not always available for such an extended time period, a solution is to use output from other atmospheric models or reanalysis data as spin-up forcing. Here, we evaluate how the choice of spin-up data set affects the subsequent model run. For this purpose, we perform two separate model experiments with a 10 year spin-up of the JSBACH model from to 2007 to 2016 using either data from the ERA5 or the MERRA-2 reanalyses as spin-up forcing. For the following model run, we use ATTO data from 2017 and 2018 as model forcing. Since the impact of the spin-up data set might vary by

season, we choose two separate starting times for the model run - one in the wet season starting in January 2017 and one in the dry season starting in September 2017.

The largest differences occur in the deeper soil layers, which is expected considering the longer adaptation time required for deeper soil layers to respond to changes in surface forcing. During the first week of the model run, soil temperature differences are $0.4\,\mathrm{K}$ in the layer between 0.07 and $0.32\,\mathrm{m}$ depth and even $1.3\,\mathrm{K}$ at about $2\,\mathrm{m}$ depth (Fig. 6a). While the temperature
differences in the upper layers decay within a few months, there is still a difference of $0.6\,\mathrm{K}$ in the deepest layer after 12 months of model run time.

For soil water, differences are also largest in the deeper layers with values exceeding $5\,\%$ between 0.32 and $1.23\,\mathrm{m}$ and of up to $20\,\%$ at about $2\,\mathrm{m}$ depth in the first month after spin-up (Fig. 6b). Differences for soil water and also for sensible and latent fluxes (Fig. 6c) decrease much quicker than for soil temperature and reach negligible values after about two months. When the
model run is started in the dry season, there is a longer impact on soil water and fluxes of about six months (not shown). This rather quick decay of the soil water differences is probably specific for the ATTO site, where a relatively shallow soil layer and large precipitation sums in the wet season often cause a saturation of the soil layers with water. Consequently, this finding cannot a priori be transferred to other sites and model grid points.

As a second step, we aim to evaluate whether the size of the biases induced by the different spin-up data sets are relevant
compared to general model biases. We quantify the model biases by comparing model runs with an optimized spin-up data set (for details see Sect. 3.2.3) to measurements from the ATTO site. The relative bias is then calculated as the fraction between spin-up bias and model bias. For both soil temperature and water (Fig. 6d,e) the choice of the spin-up data set has a larger impact in the wet season than in the dry season. For soil temperature, values are larger in the layer around $0.7\,\mathrm{m}$ depth with spin-up biases in the wet season amounting to up to $75\,\%$ of the model bias in the first month after spin-up and of still $20\,\%$
after seven months. In the layer above, at around $0.2\,\mathrm{m}$, spin-up biases are only relevant in the first month. Model biases for soil water in the wet season are slightly distorted since the soil is often saturated with water (see also Sect. 3.2.3). Results are more meaningful in the dry season, where spin-up biases amount to up to $25\,\%$ of the observed model biases.

As a next step, we analyze the impact of the choice of forcing data set on variables associated with plant growth. Figure 7a and b show the differences observed in gross and net primary productivity (GPP and NPP), respectively. Since the diurnal
cycles of these variables contribute significantly to their overall variability, we focus on cumulative values to minimize the impact of these cycles. The cumulated differences amount to more than $0.1\,\mathrm{g\,m^{-2}}$ for GPP, accounting for about $1.5\,\%$ of the average annual sums. For NPP, the cumulative differences vary depending on the starting time of the model run. For instance, when the run starts during the dry season (S2), the differences are substantially larger, exceeding $4\,\%$ of the average annual sums, while for the run started in the wet season (S1), the differences are less than $1\,\%$. This discrepancy can be attributed to
the fact that the differences persist twice as long for S2 compared to S1.

The differences in canopy conductance (Fig. 7c), a parameter associated with photosynthesis and transpiration, reinforce the same conclusions. The largest differences for canopy conductance occur for S1 within the first month, which is also the time of the largest soil water differences (Fig. 6). The MERRA-2 spin-up leads to a drier soil, which subsequently restricts stomatal opening and thereby limits the rate of photosynthesis and transpiration. Consequently, the reduced photosynthesis results in a

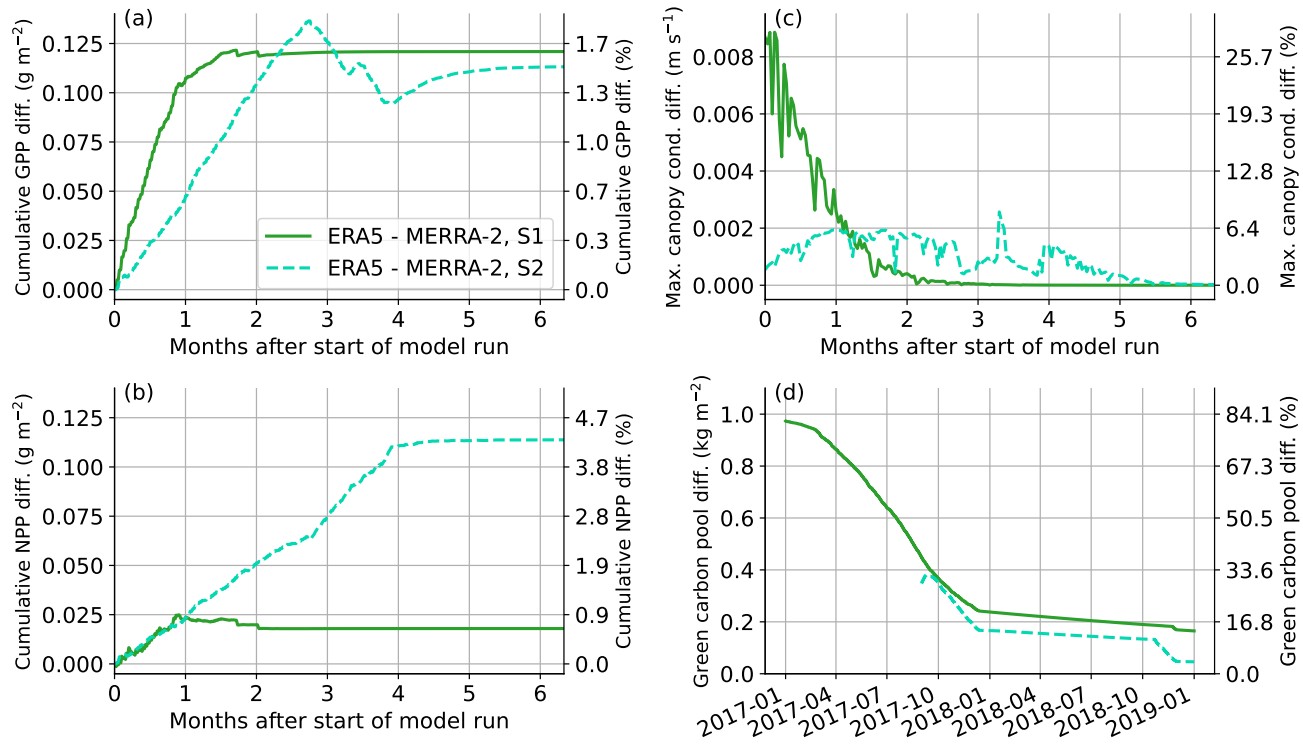

**Figure 7.** Consequences of different spin-up data sets on plant activity: model results using ERA5 data for spin-up minus those using MERRA-2. Solid lines represent the model run started in January 2017 (S1) and dashed lines the one started in September 2017 (S2). Absolute differences (left y-axes) are shown for the cumulative differences of gross primary production (a), net primary production (b), the daily maximum differences of the canopy conductivity (c) and the green carbon pool (d). The right y-axes represent relative differences with respect to average annual sums of GPP (a) and NPP (b), and the overall maximum of modeled canopy conductivity (c) and the green carbon pool (d) in the years 2017 and 2018.

smaller green carbon pool for the MERRA-2 spin-up (Fig. 7d). In the first two months after the start of the model run in January 2017, the differences in the green carbon pool amount to more than 80% of the annual average, with values remaining above 10% even after two years for S1. On the other hand, changes in the wood carbon pool occur over much longer time scales and may not reach equilibrium even after a 10-year spin-up and therefore the results should be interpreted with care. It is worth noting that the choice of spin-up data set causes differences in the order of 5 % of the annual means, and these differences only

slightly decrease throughout the two years of the model run (not shown).

Taken together, these findings highlight that the choice of the spin-up data set significantly impacts the model results, particularly for shorter model runs spanning only a few days or weeks. This influence cannot be overlooked and should be considered when setting up a simulation with a land surface model.

### 3.2.2 Sensitivity to wind speed and precipitation biases

The comparisons in Sect. 3.1 revealed significant biases between reanalysis data and ATTO measurements. Here, we quantify the impact of these biases on the JSBACH model results when reanalysis data are used as model forcing. We found the largest annual mean biases for MERRA-2 wind speed and ERA5 precipitation. For the sensitivity tests, we set up a JSBACH model run with a 10-year spin-up period and a model run of one year in 2017. The results presented in Sect. 3.1.1 indicate that the underestimation of the wind speed by the two reanalyses is a complex issue. For simplicity, we apply a very simple bias

correction in this sensitivity study by adding an offset of the annual mean wind speed bias of -1.2 $\mathrm{ms}^{-1}$ between 2014 and 2018 to the MERRA-2 data. The results are then compared to those using the original MERRA-2 forcing. In a second sensitivity study, we use a factor to adjust the ERA5 maximum annual precipitation bias of 44 % and compare the results to those using original ERA5 forcing. Figure 8 shows the monthly mean differences between the "original" and the "adjusted" model results.

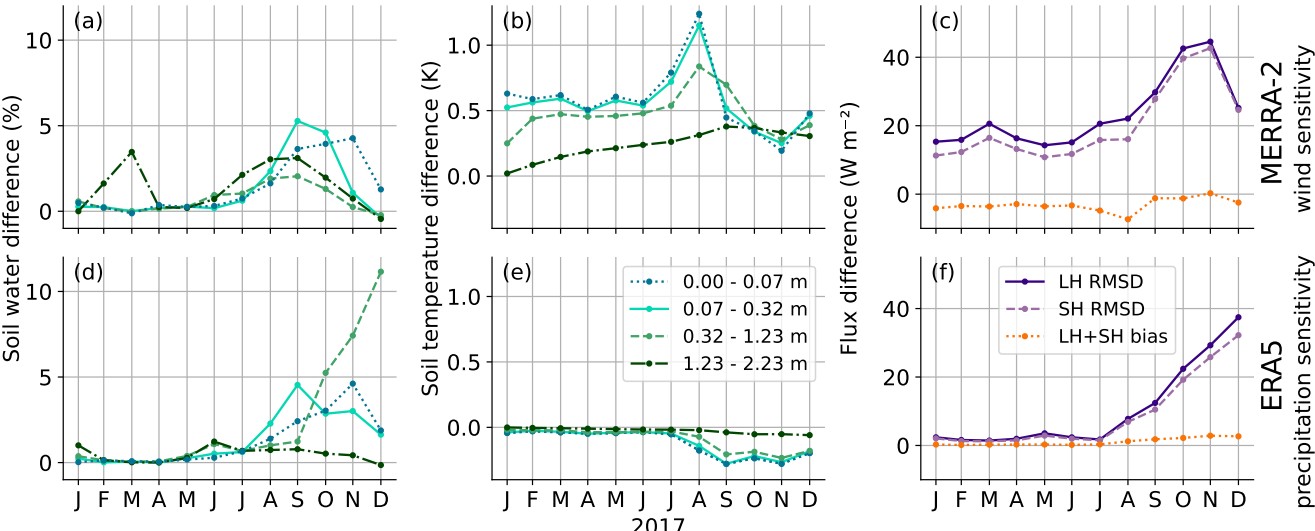

**Figure 8.** Monthly mean differences between the original and the adjusted JSBACH model runs for the MERRA-2 wind (top) and ERA5 precipitation (bottom) sensitivity runs. Differences for soil water content (a,d) and soil temperatures (b,e) are presented for different vertical soil layers of the model. Panels (c) and (f) show root mean squared differences (RMSD) for sensible (SH) and latent (LH) heat fluxes and biases of net turbulent fluxes.

The wind sensitivity runs show the largest soil water differences in the dry season. In the original run, monthly mean

water content in all depths is overestimated by up to 5 % (Fig. 8a) and in the upper two layers daily mean soil water content is overestimated by up to 10 % (not shown). Soil temperatures of the original run are also slightly too high in the dry season (Fig. 8b). These differences are also related to the turbulent exchange at the surface. Correction of the wind speed bias results in an increase of the forcing wind speed in the adjusted run, which causes an increase of the absolute value of both turbulent latent and sensible heat fluxes. However, in the dry season these flux changes counteract each other and the resulting net turbulent flux

bias is close to zero (Fig. 8c). In the wet season, the largest differences occur for soil temperatures down to a depth of 1.23 m. The original run overestimates monthly mean soil temperatures by about 0.5 K (Fig. 8b) and daily mean temperatures by up to 1.2 K in the upper two layers (not shown). Other than in the dry season, an increased wind speed in the adjusted run an the resulting stronger evaporation does not decrease the soil water content since the soil is mostly saturated with water in the wet season. As a result, changed latent and sensible heat fluxes do not fully counteract each other resulting in an underestimation

of the net turbulent heat fluxes by about $4\,\mathrm{W\,m}^{-2}$ by the original run.

       For the precipitation sensitivity run, all changes in the wet season are negligible. This means that - even by decreasing precipitation sums by 44 % in the adjusted run - the soil is still mostly saturated with water. Changes are more notable in the dry season. The original run overestimates the monthly mean soil water content in the upper two layers by up to 5 % (Fig. 8d), which is the same order of magnitude as in the wind sensitivity run. An overestimated soil water content results in increased

latent heat fluxes, which in turn cool the surface. This is evident from an underestimation of the soil temperatures in the upper three layers by up to 0.3 K (Fig. 8e) and also from the differences of the net turbulent heat fluxes in the dry season by about $2\,\mathrm{W\,m}^{-2}$. Overall, these sensitivity studies demonstrate that biases of the forcing data sets can have a non-negligible impact on the model results - at the ATTO site especially in the dry season - and should be corrected if possible.

### 3.2.3    Comparison of model results to ATTO measurements

In this section, we compare ATTO measurements to model results to identify possible shortcomings of the JSBACH land surface model. Based on the results of the previous sections we construct an optimized version of the spin-up run. The findings from Sect. 3.2.1 indicate that the duration during which the choice of spin-up forcing data set has a significant impact on most variables is less than one year. As a result, a spin-up period of two or three years would be sufficient to reach an equilibrium state for soil water content and soil temperatures in the upper layers at this specific site. However, variables like temperatures

of the deeper soil layers or the green carbon pool require a longer spin-up duration. Therefore, when employing a standalone land surface model, the selection of the spin-up period should be determined by the specific processes of interest. In our case, we adopt a cautious approach and use a 10-year spin-up period for the model, which has the following characteristics: We use ERA5 because for most forcing variables the shape of the annual cycles agree better with measurements. Wind speed bias ($-0.8\,\mathrm{m\,s}^{-1}$) and mean precipitation bias (+32%) are corrected and 9 years of model spin-up are performed (2007-2015).

The last spin-up year is performed using ATTO data. Two years of model results forced by ATTO data are then compared to measurements. By using this setup, we can minimize the impact of the choice of spin-up forcing data set on the results and thus focus mainly on uncertainties caused by model physics. For the comparisons, missing values of the measurement data are also flagged in the model results. We evaluate the model performance based on average annual cycles of soil water and temperature and seasonally averaged soil profiles and turbulent fluxes.

ATTO soil measurements are only available for certain model soil layers (see also Tab. A1). Therefore, we compare model results from the second (0.07 to 0.32 m) and third (0.32 - 1.23 m) soil layers and average the corresponding ATTO measurements, which are 0.1 to 0.3 m and 0.4 to 1.0 m, respectively. Please note that soil water measurements reach deeper than soil temperature measurements, which are only available at 0.1, 0.2 and 0.4 m. Figure 9 shows annual cycles of soil water content

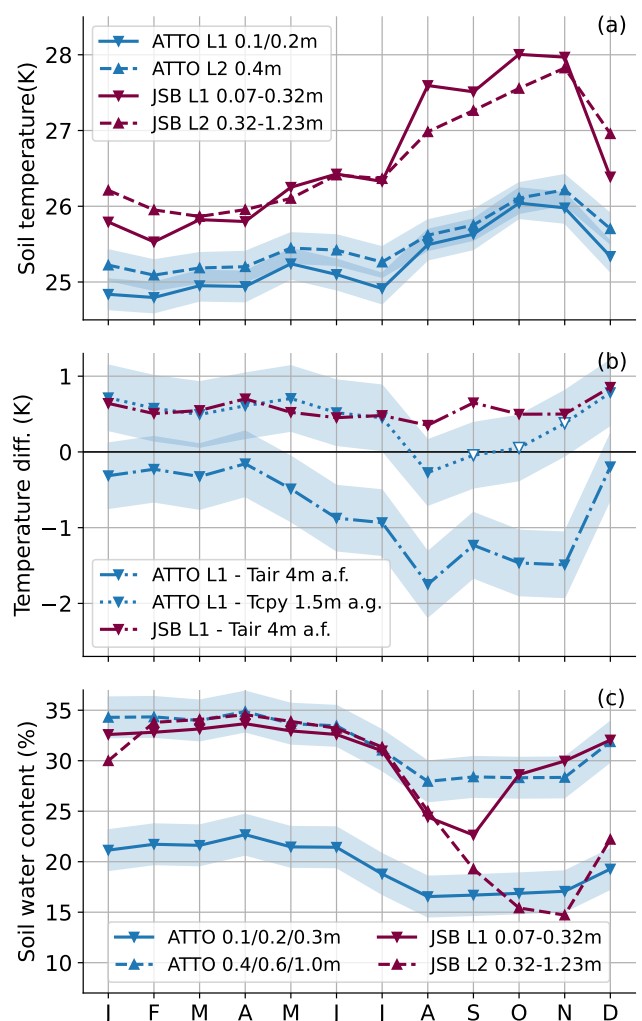

**Figure 9.** Annual cycle of soil temperatures (a) averaged over the years 2017 and 2018 from ATTO measurements and the JSBACH model. Solid lines represent the second and dashed lines the third model soil layer and the corresponding ATTO measurements height, respectively. (b) Annual cycles of temperature differences between soil layers L1 in (a) and air temperatures measured about 4 m above the forest (a.f.) or within the forest canopy at 1.5 m above the ground (a.g.). Empty symbols represent months with more than 30 % of missing data. (c) As in panel (a), but for soil water content. Shading in all panels denotes the measurement uncertainty.

and temperature averaged over the two years of the model run from 2017 to 2018. It is evident that soil temperatures are over-
estimated in the model in all months and for both considered depths (Fig. 9a) with large annual mean biases of -1.1 K in the
upper and -1.3 K in the lower layer.

To identify possible reasons for this bias we analyzed temperature differences between the soil and the air above. The land surface model is forced by air temperatures measured roughly 4 m above the forest canopy. It is evident from Fig. 9b that soil temperatures in the model are always higher than air temperatures with a difference of about 0.6 K, which remains almost constant throughout the year. In contrast, measured soil temperatures are always lower than air temperatures 4 m above the forest with larger differences in the dry season. However, in reality soil temperature depend more likely on air temperatures close to the ground within the canopy layer than on temperatures above the forest. We thus also analyzed differences between ATTO soil temperatures and air temperatures measured at 1.5 m above the ground. Here, we find a much better agreement with the modeled temperature differences throughout the year and even an almost perfect match in the wet season. This indicates that the observed bias between measured and modeled soil temperatures is likely caused by a cooling effect within the canopy layer, which is not included in the land surface model - such as evaporation of rain from stems and leaves.

In contrast, differences between modeled and measured soil water content show larger seasonal dependence (Fig. 9c). In the upper layer, the model overestimates the soil water content with an almost constant monthly bias of about 11 %. In the lower layer, modeled soil water agrees very well with measurements in the wetter season from February to July. However, in the dry months from September to December the model underestimates soil water content - also by about 11 %. The fact that different soil layers exhibit different bias signs hints towards problems of vertical soil water transport in the model. To analyze vertical dynamics more deeply we look at soil profiles in the following.

Figure 10 shows boxplots of daily mean soil temperature and water content for different depths. For soil water content, ATTO measurements show very similar profile shapes for the wet and dry seasons, with lower values in the upper soil layers and higher values in deeper layers. The minimum and maximum can be found at 0.2 m and 0.6 m depth, respectively. However, the JSBACH model produces different profiles in the wet and dry seasons. In the wet season, all soil layers approach a constant value of about 35 % (Fig. 10a), which is close to the field capacity of the soil. This indicates that the soil water content approaches saturation levels during rainfall events. In the dry season, the maximum soil water content is found in the uppermost two layers until 0.32 m, which is contrasting the measurements, for which a minimum is found in these depths (Fig. 10b).

A possible reason for the discrepancies of the soil water profiles between model and measurements in both seasons could be an overestimation of the soil water storage capacity in the upper two layers. This could either mean that boundary conditions of the soil properties do not represent the actual conditions at ATTO properly. Or - since JSBACH prescribes the same soil properties in all model layers - the soil at the ATTO site might actually be heterogeneous and consist of different soil textures in different depths. Another relevant process could be transpiration, which is a sink term in the soil water balance equation and is constant with depth in JSBACH. However, it is well known that root density - and thus water extraction from the soil by plants - varies with depth (e.g. Feddes, 1982; Perrochet, 1987). Improving the representation of roots in JSBACH (based on common approaches used in other land surface models, see e.g. Zheng and Wang (2007)) would potentially change the soil water profile, which we suggest should be investigated further in the future. Other studies indicated that it is beneficial to adopt an exponential root profile assumption (e.g. Jackson et al., 1996; Zeng, 2001).

Profiles of soil temperature also show larger seasonal differences in the model results than in the observations. In the wet season, the shapes of the profiles agree reasonably well, with slightly increasing temperatures with depth both in the model

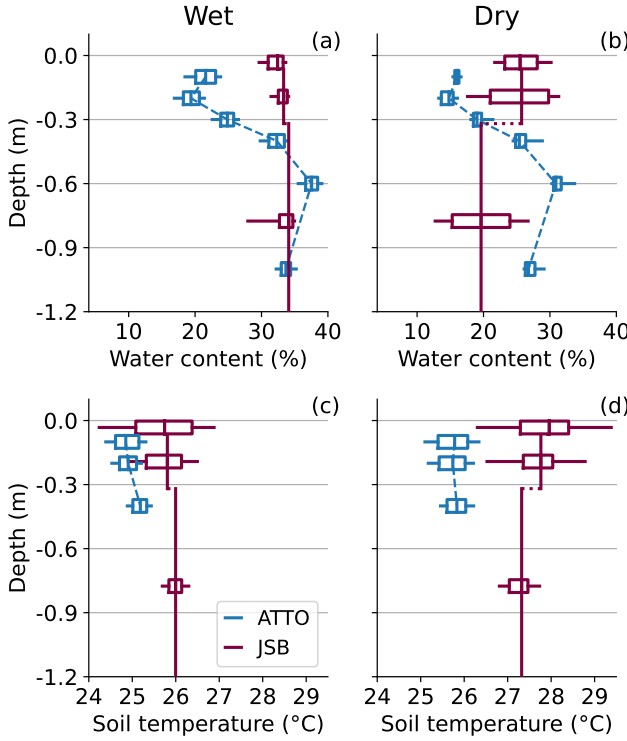

**Figure 10.** Vertical profiles of soil water content (a,b) and soil temperatures (c,d) during the wet (left) and dry seasons (right) in 2017 and 2018. Vertical lines denote the median values of daily means for the respective heights and boxplots - showing quartiles (box) and 10 %- and 90 %-percentiles (whiskers) - are centered at the ATTO measurement depths (blue) and the center of the JSBACH model layer (purple).

and observations (Fig. 10c) and biases below $1 \, \text{K}$. However, in the dry season, model temperatures are higher in the upper two layers until $0.32 \, \text{m}$ than below, while the measured temperature profile has on average nearly constant temperatures (Fig. 10d) with biases exceeding $1.5 \, \text{K}$ in all depths. The whiskers of the boxplots (10 % and 90 % percentiles) also indicate that the day-to-day variability is overestimated in the model.

To get more insight into the temperature variability with depth we look at average diurnal cycles of soil temperature. Since the changes of the overall characteristics with seasons are small, we only present annual mean diurnal cycles in Fig. 11a. We focus on model layer 2 between 0.07 and $0.32 \, \text{m}$ for which observation depths show the best match. We would expect the best agreement with the observations at $0.2 \, \text{m}$ depth. However, the amplitude of the diurnal cycle is strongly overestimated and the model agrees better with the amplitude measured at $0.1 \, \text{m}$ depth (Fig. 11b).

Typically, diurnal and annual soil temperatures have a sinusoidal shape with decreasing amplitude with depth and a phase shift, both depending on the heat conductivity of the soil. Our results indicate that the penetration depth of the diurnal cycle is overestimated in the model, which might be related to an overestimation of the heat conductivity of the soil. This is also in line with our findings of an overestimated soil water content (see above) since wet soil has a larger heat conductivity than dry soil.

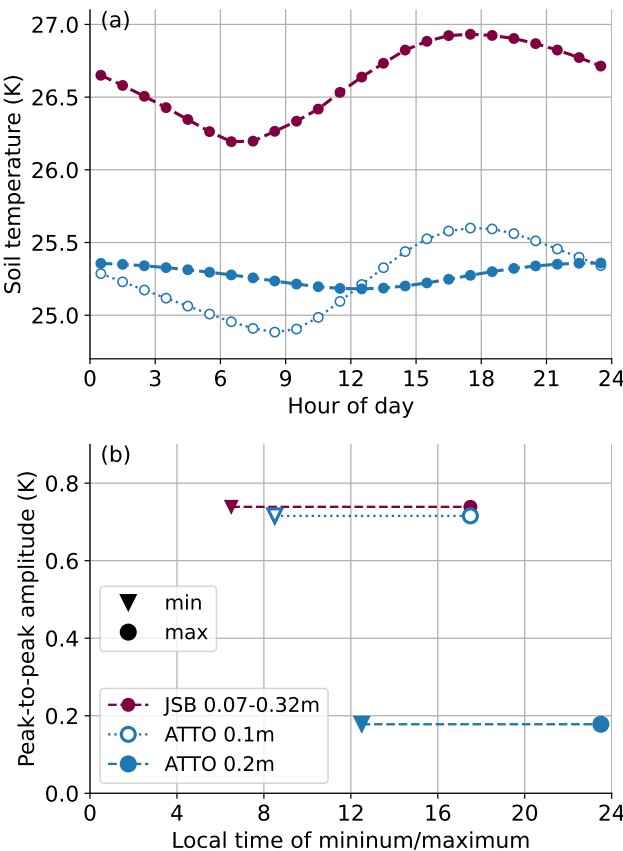

**Figure 11.** (a) Diurnal cycle of soil temperatures averaged over the years 2017 and 2018 in the second JSBACH model layer and the corresponding ATTO measurement depths. (b) Corresponding amplitudes and times of minimum and maximum of the curves in (a).

To further test this hypothesis, we make use of the formula describing the exponentially decreasing soil temperature amplitude $A$ with depth (see e.g. Moene and Dam, 2014):

$$A(z) = A_0 \exp\left(-\frac{z}{d}\right), \tag{1}$$

where $A_0$ is the amplitude at the surface and $d$ is the damping depth, which is a function of the heat conductivity of the soil. Using the amplitudes of the diurnal temperature variations from the two uppermost layers $z_1$ and $z_2$, we can determine the

damping depth as:

$$d = \frac{z_2 - z_1}{\ln\left(A(z_1)/A(z_2)\right)}. \tag{2}$$

For both the ATTO measurements (at 0.1 and 0.2 m) and the model results (layers centered at 0.03 and 0.19 m) we obtain a damping depth of about 7 cm. This good agreement indicates that properties concerning heat transport in the soil are well represented in the model, which counteracts the above hypothesis.

The overestimation of the penetration depth could also be related to the amplitude of the temperature at the soil surface $A_0$. Using the value of $d = 0.07\,\mathrm{m}$, we obtain $A_0 = 3.0\,\mathrm{K}$ using ATTO measurements at $0.1\,\mathrm{m}$ depth but a much larger value of $A_0 = 8.5\,\mathrm{K}$ using model results from the uppermost model layer. This gives a hint that the forest canopy dampens the soil surface temperature. However, JSBACH (version 4) does not include an explicit canopy layer or a parametrization of the canopy heat storage effect. Consequently, the model is not able to capture this dampening effect.

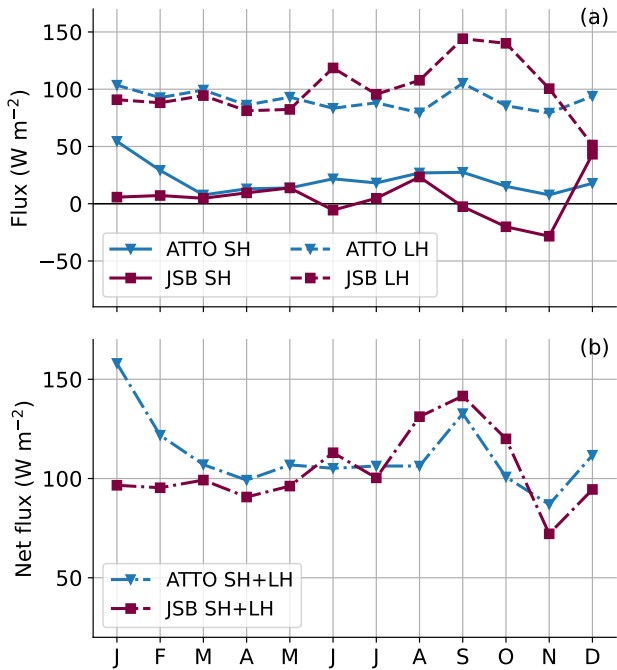

**Figure 12.** Annual cycles of turbulent sensible (SH) and latent (LH) heat fluxes (a) and net fluxes (b) averaged over the years 2017 and 2018 from ATTO measurements and the JSBACH model. A positive sign denotes upward fluxes from the soil to the atmosphere.

As a final step, we compared measured turbulent heat fluxes to those from the model. Two major aspects had to be considered before the comparison. First, time series of measured turbulent fluxes contain many gaps - mostly due to rain - and in 2017 and 2018 there is not a single day with full data coverage. Since the model is forced with ATTO measurements, rain occurs at the same time in the model and in reality and those cases can therefore be masked simultaneously. It needs to be kept in mind that data availability is much lower during the day (30 to 40 % at midday) than at night (about 80 %) and thus the diurnal cycles should be interpreted with care. To obtain more representative monthly means we first calculated average diurnal cycles for each month, which were then averaged to obtain monthly values. By doing so, we reduce the impact of a higher data availability at night on the overall monthly values. The second issue concerns the height differences between the ATTO measurements at about $44\,\mathrm{m}$ above the forest and the model results at the surface. In a well mixed atmospheric boundary layer, turbulent fluxes typically decrease linearly with height with a zero crossing at the height of the boundary layer depth. To get a rough estimate, we

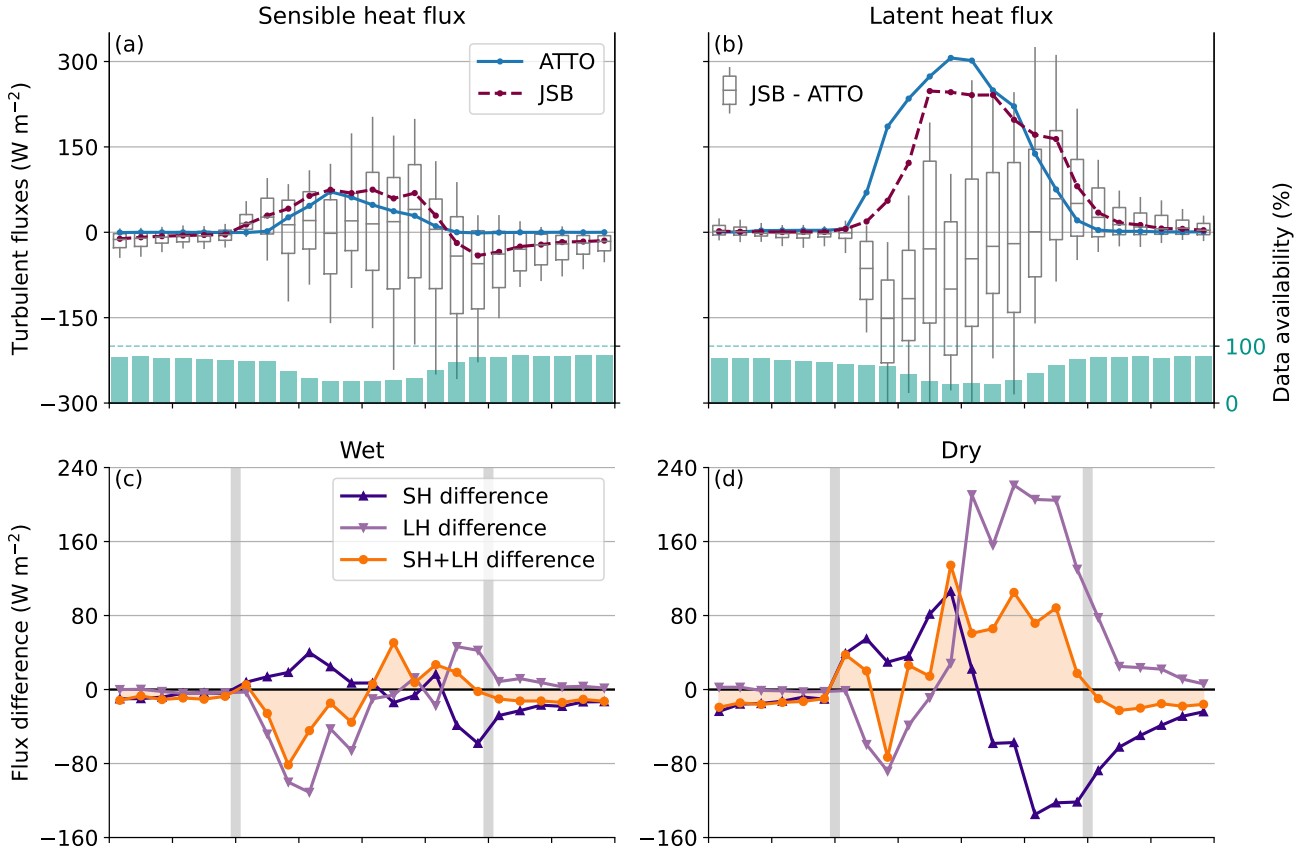

**Figure 13.** Top: Annual median diurnal cycles of sensible (a) and latent (b) heat fluxes from the JSBACH model and measured at ATTO in 2017 and 2018. Boxplots display differences between the model and the measurements with quartiles as box and 10 %- and 90 %-percentiles as whiskers. Light blue bars indicate the fraction of available measurements. Bottom: Median diurnal cycles of flux differences between model and measurements for sensible and latent heat flux and their sum for the wet (c) and dry (d) seasons. Gray vertical lines mark the times of sunrise and sunset, respectively.

calculated a height correction factor for the measurements using boundary layer heights from ERA5 and MERRA-2. However, this correction only had a notable impact on a few days but did not change the overall conclusions drawn from the analysis of flux differences between model and measurements and thus, the corrected results are not shown here.

Figure 12 shows average annual cycles of sensible (SH) and latent (LH) heat fluxes and the sums of the two. It is evident that measured LH fluxes are by a factor of about three to four larger than SH fluxes. Generally, modeled and measured fluxes agree quite well. In the dry season, LH fluxes are slightly overestimated by the model, which is however counteracted by
underestimated SH fluxes and thus net fluxes show a good agreement.

Diurnal cycles of turbulent heat fluxes are presented in Fig. 13. It is notable that LH fluxes in the model show a time shift compared with the measurements with fluxes increasing and decreasing about 2 to 3 hours later in the day (Fig. 13b). On the one hand, this results in an underestimation of LH fluxes in the morning of up to $100\,\mathrm{W\,m^{-2}}$. As a consequence, the surface warms up too quickly causing an overestimation of the SH fluxes - which does, however, not fully compensate for the overestimation of LH fluxes, resulting in a net bias of about -50 to $-100\,\mathrm{W\,m^{-2}}$. On the other hand, LH fluxes are overestimated in the afternoon with the largest values during the hours before sunset. The impact is most notable in the dry season (Fig. 13d). Median LH fluxes are overestimated by more than $200\,\mathrm{W\,m^{-2}}$ in the model, which leads to an increased cooling of the surface and resulting decreasing SH fluxes. Net turbulent fluxes are overestimated by about $100\,\mathrm{W\,m^{-2}}$ in the afternoon. Due to the considerable amount of missing measurements, however, it remains difficult to determine whether these biases have a significant impact on longer time periods.

To summarize, our comparison of JSBACH model results with soil and surface measurements at the ATTO site indicates that the following model processes should be evaluated more carefully in the future: Firstly, the model performance could possibly be improved by a better representation of the vertical structure of the soil. It would be useful to allow for different soil textures in different model layers and additionally, more detailed knowledge of soil properties - either from field measurements at the considered site, or from high resolution data sets of soil textures - could be used as boundary conditions. Furthermore, the vertical distribution of the root density should be considered to allow for a varying soil water sink with depth due to transpiration. Secondly, it would be beneficial to include a representation of the canopy heat storage effect into the model. This could be accomplished by modeling the processes in a separate canopy layer explicitly or by adopting a simpler approach that parametrizes the heat storage by the canopy. Heidkamp et al. (2018) and Schulz and Vogel (2020) demonstrated that a simple approach, which is based on a skin temperature formulation, reduces the underestimation of the amplitude of the diurnal cycle of surface and soil temperatures and the corresponding incorrect phase shifts. Moreover, the skin temperature formulation improves biases in latent and sensible heat fluxes (Schulz and Vogel, 2020; Renner et al., 2021). To reduce the soil temperature bias of the model, it might also be beneficial to re-evaluate the representation of additional cooling terms within the canopy layer. For example, evaporation of dew or of rainfall intercepted by the vegetation impacts near-surface air temperatures, which then in turn influence soil temperatures.

## 4   Summary and conclusions

In this study we used the land surface model JSBACH in a site-level setup to study its performance at a rainforest site - hereby focusing especially on processes influenced by the forest canopy. Observations at the Amazon Tall Tower Observatory were used to evaluate the model performance and, as a first step, to optimize the external forcing data of the land surface model.

First, we compared near-surface atmospheric variables from ERA5 and MERRA-2 to five years of ATTO measurements to determine whether reanalysis data are generally suitable to be used as forcing data for the land surface model. For wind, both reanalyses were able to simulate the shape of the diurnal cycle correctly but strongly underestimated wind speeds at all heights

by about $1\,\mathrm{m\,s^{-1}}$. Biases of the same order of magnitude have been found in other studies, which also report that the sign of the bias varies with region (Jourdier, 2020; Staffell and Pfenninger, 2016; Carvalho, 2019) and terrain (Gualtieri, 2021).

MERRA-2 reproduced the annual cycle of precipitation quite well, while ERA5 overestimated annual mean precipitation by 30 %. Since the amount of incoming shortwave radiation was also underestimated, it is likely that the cloud cover in ERA5 was too large. ATTO measurements showed a bimodal distribution of diurnal precipitation with a second peak around the time of sun rise. ERA5 also produced early-morning precipitation but too early at night and MERRA-2 completely failed to capture the second peak.

Contrary to precipitation, ERA5 captured the annual cycle of temperature better than MERRA-2. In the dry season, MERRA-2 overestimated monthly mean temperatures by more than $1\,\mathrm{K}$, which is related to too large daily maximum temperatures in this season. For specific humidity, both reanalyses did not reproduce the shape of the diurnal cycle correctly. While ATTO measurements showed increasing humidity during the morning with a maximum around mid-day, both reanalyses produce a local minimum of humidity during this time, which is possibly caused by an overestimated growth rate of the atmospheric boundary layer.

Next, we tested how much these biases of the reanalysis data affect the results of the land surface model. Comparing different model simulations, which use either ERA5 or MERRA-2 as forcing for the spin-up period, we found the largest differences of up to 20 % for soil water content and up to $1.3\,\mathrm{K}$ for soil temperatures in the deepest soil layers. The differences for deeper soil temperatures and the green carbon pool remained non-negligible even after one year. For other variables associated with plant growth (GPP, NPP, canopy conductance), noticeable differences were observed for up to six months after the start of the model run. For both soil temperature and water the choice of the spin-up data set accounted for more than 10 % of the observed model biases during the first three months after spin-up in $0.7\,\mathrm{m}$ depth and for more than 50 % closer to the surface. Thus - especially for shorter model runs of only a few days or weeks - the choice of spin-up data set is not negligible and can have a large impact on the model results.

Correcting biases of the forcing data also changes the land surface model results. We conducted sensitivity runs for wind speed and precipitation and found the largest changes in the dry season, when the soil is not saturated with water. For both variables, soil water differences amounted to up to 5 % for monthly means and 10 % for daily means in all depths. Too low wind speeds caused an overestimation of soil temperatures by about $0.5\,\mathrm{K}$ in all months. These results indicate that biases of the forcing data sets can have a notable impact on the model results and should be checked and corrected beforehand if possible.

Based on these results, we performed a model run with optimized forcing. The spin-up consists of nine years of ERA5 data (with corrected wind speed and precipitation biases), followed by one year of ATTO data. Results from a two-year model run forced by ATTO measurements were then compared to the observations of soil water content, soil temperatures and turbulent heat fluxes to identify possible model shortcomings. Comparing profiles of soil water we found that the model generally overestimates the water content in the upper layers until $0.32\,\mathrm{m}$. On the one hand, this could be related to the choice of boundary conditions, like the overall soil type. It could also be caused by a lack of vertical differences of soil textures or of the root density, which impacts transpiration.

We also found that the model overestimates soil temperatures by about $1\,\mathrm{K}$ with a slightly larger bias in the dry season.
Comparisons with measured air temperatures above and within the canopy suggest that this is most likely caused by additional cooling within the canopy layer, e.g. by evaporation of rainfall intercepted by the vegetation, which is not sufficiently accounted for in the model. Furthermore, the penetration depth of the diurnal cycle of temperatures is overestimated. A more detailed analysis of the change of temperature amplitude with depth indicates that this is likely related to the temperature dampening effect of the forest canopy, which has not been incorporated into the JSBACH model, yet. The analysis of turbulent fluxes revealed a timing mismatch of the diurnal cycles of latent and sensible heat fluxes. ATTO flux measurements contain a large amount of missing data, however, and thus further analysis of longer and more complete time series would be required to examine the underlying causes in more detail.

To conclude, we suggest that future improvements of the JSBACH model could possibly focus on allowing more vertical variability - both of soil texture and root density - within the soil column. Furthermore, a separate canopy layer would likely improve processes related to the energy transport within the soil. When implementing and evaluating future improvements of the canopy scheme, we suggest to consider especially the bias and phase shift of the soil temperature. Finally, we showed that the ATTO site provides an ideal framework for testing canopy related processes in LSMs and will certainly be useful in future modeling studies.

## Appendix A:  Meteorological instrumentation at the ATTO site

In this study, we used measurements of various meteorological variables from the ATTO site. Details about instruments, measurement heights and measurement uncertainties are presented in Tab. A1.

## Appendix B:  Humidity conversion

For comparison of the humidity with reanalysis data we convert the relative humidity $RH$ to specific humidity $q$ with

$$q = \frac{0.622\,\mathrm{Pa}}{p}\left(\frac{RH}{100\,\%}\mathrm{e}_{\mathrm{sat}}(\vartheta)\right), \tag{B1}$$

where $p$ is the air pressure in $\mathrm{Pa}$ and $\mathrm{e}_{\mathrm{sat}}$ is the saturation water vapor pressure calculated using the Magnus equation with $\vartheta$ being the air temperature in $^\circ\mathrm{C}$. Since air pressure is only measured at $81\,\mathrm{m}$ height, we extrapolate the values downward to the measurement heights of humidity using the barometric formula:

$$p_2 = p_1 \cdot \exp\left(-\frac{g\,(z_1 - z_2)}{\mathrm{R}_\mathrm{l} T_v}\right). \tag{B2}$$

$p_i$ is the pressure at height $z_i$, $g = 9.81\,\mathrm{ms}^{-1}$ is the gravity of Earth, $R_l = 287.1\,\mathrm{Jkg}^{-1}\mathrm{K}^{-1}$ is the specific gas constant for dry air and the virtual temperature is calculated as $T_v = T(1 + 0.61q)$. Air pressure is calculated stepwise downwards and if data required for the calculations are missing at the considered height, values are replaced by applying Eq. B2 to the levels above.

**Table A1.** Observations at the ATTO site used in this study.

| Variables | Instruments | Measurement heights (m) | Accuracy |
|---|---|---|---|
| Air temperature and relative humidity | Termohygrometer (CS215, Rotronic Measurement Solutions,UK) | 1.5, 36.3, 40.2, 55.3 | Temperature: $0.3\,°C$ (at $25\,°C$), $0.4\,°C$ (5 to $40\,°C$); relative humidity: $4\,\%$ (0-100 % range), $2\,\%$ (10-90 % range) at $25\,°C$ |
| Wind speed and direction | 2-D sonic anemometer (WindSonic, Gill Instruments Ltd., UK) | 43.1*, 50.8, 66.0, 73.7 | Wind speed uncertainty: $2\,\%$ at $12\,\mathrm{m\,s^{-1}}$ |
| Turbulent sensible and latent heat flux | **Wind, temperature:** 2014-2017: 3D Sonic Anemometer (WindMaster, Gill Instruments Ltd., UK); 2017-2018: 3D Sonic Anemometer (CSAT3, Campbell Scientific, Inc., USA) **Humidity:** Infrared Gas Analyzer (IRGA LI-7500A, LI-COR Inc., USA) | 82.0 | **WindMaster:** wind speed: $<1.5\,\%$ RMS at $12\,\mathrm{m\,s^{-1}}$ **CSAT3:** wind speed $<2\,\%$ (wind vector within $\pm 5°$ of horizontal) temperature: $0.01\,\mathrm{K}$ $H_2O$ content: $<2\,\%$ |
| Precipitation | Rain gauge (TB4, Hydrological Services Pty. Ltd., Australia) | 81.3 | Rainfall per tip: $0.254\,\mathrm{mm}$; uncertainty: $2\,\%$ ($<250\,\mathrm{mm\,h^{-1}}$), $3\,\%$ (250-500 $\mathrm{mm\,h^{-1}}$) |
| Atmospheric pressure | Barometer (PTB101B, Vaisala, Finnland) | 81.0 | $0.5\,\mathrm{hPa}$ at $20\,°C$; $1.5\,\mathrm{hPa}$ (0-40 $°C$) |
| Shortwave radiation | Pyranometer (CMP21,Kipp & Zonen, Netherlands) | 75.6 | Expected daily uncertainty $<2\,\%$ |
| Longwave radiation | Pyrgeometer (CGR4, Kipp & Zonen, Netherlands) | 75.6 | Uncertainty $<3\,\%$ for daily totals |
| Soil water content | Water content reflectometer (CS615, Campbell Scientific Inc.,USA) | -0.1, -0.2, -0.3, -0.4, -0.6, -1.0 | $2\,\%$ manufacturers standard calibration |
| Soil temperature | Thermistor (108, Campbell Scientific Inc., USA) | -0.1, -0.2, -0.4 | $0.2\,\mathrm{K}$ |

* wind measurements at 43.1 m height are only available for the years 2014 and 2015

## Appendix C: Early morning precipitation

To study the regional distribution of average precipitation patterns we use the IMERG data set (Integrated Multi-satellitE REtrievals for GPM from the NASA-JAXA Global Precipitation Measurement mission; Huffman et al. (2019)) as reference.

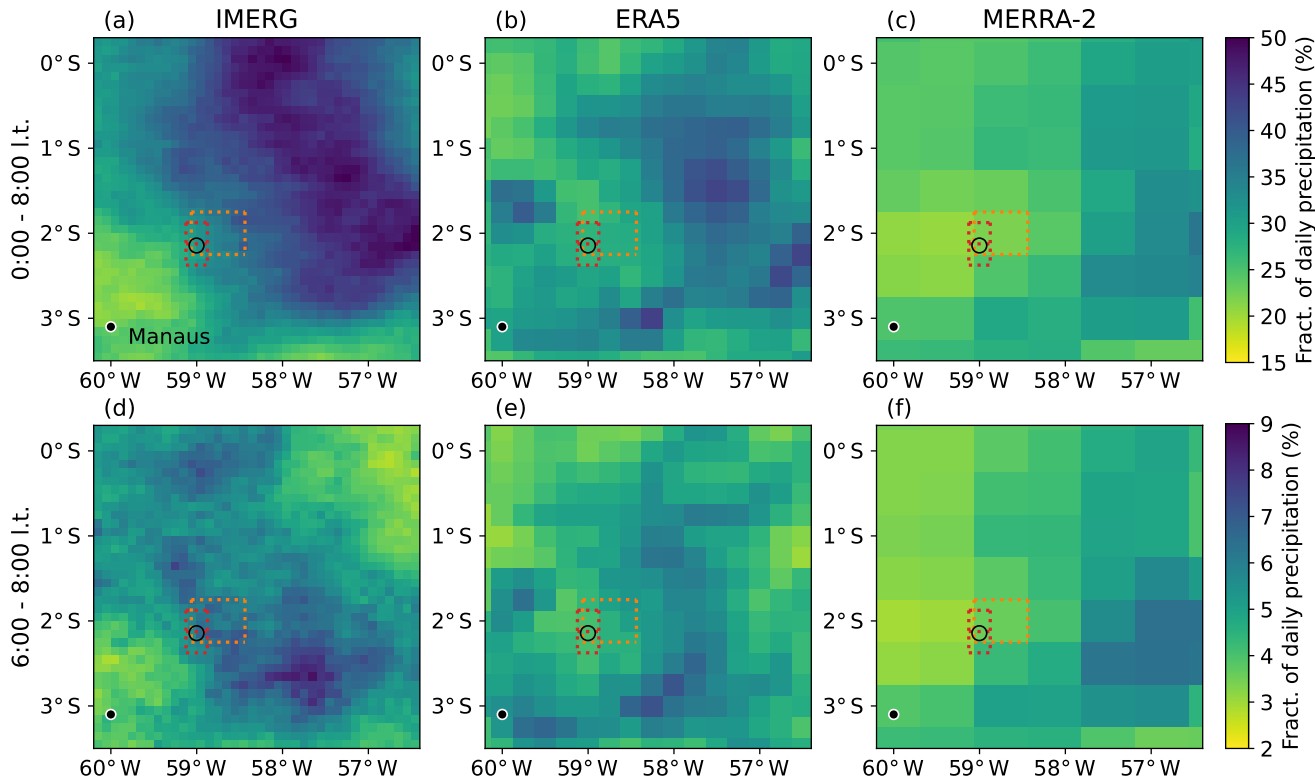

**Figure C1.** Fraction of nighttime precipitation between (top) 0:00 and 8:00 and (bottom) 6:00 and 8:00 local time averaged over the wet season (JFMA) from 2014 to 2018 based on the IMERG precipitation data set (a,d), MERRA-2 (b,e) and ERA5 (c,f) data. The circle marks the ATTO location. Red and orange boxes denote the considered grid boxes for ERA5 and MERRA-2, respectively (see also Fig. 1).

The data set provides half-hourly estimates of surface precipitation rates on a 0.1° grid. More details can be found in Tan et al. (2019). We calculate the wet season (JFMA) average early morning precipitation sums for the years 2014 to 2018 based on IMERG data and compare them to results from ERA5 and MERRA-2. The top row in Fig. C1 shows late night-early morning (LN-EM) precipitation sums for the period between 0:00 and 8:00 local time and the bottom row shows early morning (EM) precipitation sums after sunrise between 6:00 and 8:00 as a fraction of total daily precipitation.

LN-EM precipitation shows strong local gradients with up to 50 % of daily precipitation falling between 0:00 and 8:00 in the region about 200 km northeast of ATTO (Fig. C1a). Northeast was also the most common wind direction for this time period with atmospheric flow coming from the 30°- 105° sector more than half of the time (not shown). The fraction of LN-EM precipitation decreases toward the southeast to only around 25 % in the region around Manaus. EM precipitation shows a band with a local maximum from northwest to southeast of about 50 km width, where precipitation after sunrise accounts for up to 610   9 % of the daily precipitation (Fig. C1d). The ATTO location receives about 30 % of its precipitation between 0:00 and 8:00 and about 6 % in the early morning. Even though the IMERG data set does not fully reproduce the early morning precipitation

maximum measured at ATTO between 7:00 and 8:00 (see Fig. 4d), it clearly shows that there is a significant amount of EM precipitation in this region with fractions changing significantly within a few hundred kilometers, i.e. within a only a few reanalysis grid cells.

While there are notable differences between LN-EM and EM precipitation (Fig. C1a and d) for IMERG, the patterns for the two reanalysis data sets stay roughly the same before and after sunrise. Fig. 4d indicates that MERRA-2 does not capture the second EM maximum of precipitation, which is observed in the ATTO measurements. It is also evident from Fig. 4c and f, that ATTO is located at an area with relatively low fractions of nighttime precipitation. Fractions are much larger at about $100\,\mathrm{km}$ further to the east. There, the maximum of the diurnal cycle of precipitation is between 7:00 and 10:00 local time (not shown),

which agrees better with the observed ATTO results. This gives a hint that MERRA-2 does not generally fail to reproduce early morning precipitation, but partly produces it at the wrong location.

*Data availability.*  Meteorological measurements from the Amazon Tall Tower Observatory are available through the ATTO data portal (www.attodata.org). ERA5 surface data were generated using Copernicus Climate Change Service information [2021] and can be obtained from the Climate Data Store (https://cds.climate.copernicus.eu/cdsapp#!/dataset/reanalysis-era5-single-levels?tab=overview). ERA5 profile

data on model levels were obtained from ECMWF's MARS tape archive. MERRA-2 are available through the Goddard Earth Sciences Data and Information Services Center (https://disc.gsfc.nasa.gov/)

*Author contributions.*  AS and FA designed the study. AA, MS and PT collected and prepared the measurement data. AS performed the data analysis, conducted the model runs, visualized the results and prepared the manuscript. FA assisted in interpreting the results and editing of the manuscript.

*Competing interests.*  The authors declare that they have no conflict of interest.

*Acknowledgements.*  We thank the two anonymous referees for their constructive criticism, which helped improve the manuscript. This research was made possible by the German-Brazilian project ATTO, supported by the German Federal Ministry of Education and Research (BMBF contracts 01LB1001A and 01LK1602E) and the Brazilian Ministério da Ciência, Tecnologia e Inovação (MCTI/FINEP contract 01.11.01248.00) as well as the Max Planck Society. We furthermore acknowledge the support by the Amazon State University (UEA), FA-

PEAM, LBA/INPA, and SDS/CEUC/RDS-Uatumã. This work was partly funded by the Deutsche Forschungsgemeinschaft (DFG, German Research Foundation) under Germany's Excellence Strategy – EXC 2037 'CLICCS - Climate, Climatic Change, and Society' – Project Number: 390683824, contribution to the Center for Earth System Research and Sustainability (CEN) of Universität Hamburg. Thanks to ICDC, CEN, University of Hamburg for data support. The results contain modified Copernicus Climate Change Service information (2014-

2018). Neither the European Commission nor ECMWF is responsible for any use that may be made of the Copernicus information or data it

contains.

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
