# Peer review of "Modeling atmosphere-land interactions at a rainforest site - a case study using Amazon Tall Tower Observatory (ATTO) measurements and reanalyis data"

_EGUsphere, 2023_

## Author Comment (AC1)

We thank the anonymous reviewer for the time and effort they invested in critically reviewing our manuscript. Please find answers to your very helpful comments and suggestions below.

**Major points**

**1)** I found it difficult to follow the results and discussion. While it is well written from a language perspective it was hard to understand which part of the text belonged to the result and the discussion. The authors refer to point 3 as results and discussion, but I am unsure why the authors did go this way. Associated with the alternating part of results and discussion there is also a lot of jumping around with the figures. For example, the first figure mentioned in the results is Figure 3c. What about 3a and b? Figure 12 is only referred to in the text after Figure 13. I can see that the authors wanted to address each climate variable consecutively, but when the variables are spread across figures it is quite hard to follow.

While we prefer to stick with the concept of a combined results and discussion section, we appreciate your concerns. In response, we have made significant efforts to enhance the readability of the text by reorganizing its structure and rewriting certain portions. The following changes were implemented:

1. The content of Fig. 3 (annual cycles of all variables) was split up and combined with the respective Figures 4-6 (old Fig. numbers). The panels of Figures 4-6 were rearranged to match the order they are cited in the text.

2. We divided the text in Sections 3.1.1 to 3.1.3 more clearly by first describing the figures and and discussing possible reasons for the observed biases at the end of the respective sections. In Sect. 3.1.3 we first describe the results for temperature and after that those for humidity, followed by a discussion of differences between observed and modeled temperature and humidity.

3. In sections 3.2.1, 3.2.2 and 3.2.3, we adjusted the order of figure panels so that they match the order of appearance in the text.

**2)** The authors find that soil temperatures and soil water content are the key differences that arise when using different forcing datasets to drive JSBACH at the ATTO site. I am wondering about all the other variables and states of JSBACH, such as GPP, NPP, stomatal conductance and carbon pools. How are they affected by the different forcings? I understand that not all model output variables should be addressed in this study, but I think it would be very interesting for the modelling community to see if some of them are also affected by the choice of forcing dataset.

We included and described additional plots of a few variables related to plant growth in Section 3.2.1, which were available for the output of our model runs:
*"As a next step, we analyze the impact of the choice of forcing data set on variables associated with plant growth. Figure 1a and b show the differences observed in gross and net primary productivity (GPP and NPP), respectively. Since the diurnal cycles of these variables contribute significantly to their overall variability, we focus on cumulative values to minimize the impact of these cycles. The cumulated differences amount to more than $0.1\,gm^{-2}$ for GPP, accounting for about 1.5 % of the average annual sums. For NPP, the cumulative differences vary depending on the starting time of the model run. For instance, when the run starts during the dry season (S2), the differences are substantially larger, exceeding 4 % of the average annual sums, while for the run started in the wet season (S1), the differences are less than 1 %. This discrepancy can be attributed to the fact that the differences persist twice as long for S2 compared to S1.*
*The differences in canopy conductance (Fig. 1c), a parameter associated with photosynthesis and transpiration, reinforce the same conclusions. The largest differences for canopy conductance*

[Figure]

Figure 1: Consequences of different spin-up data sets on plant activity: model results using ERA5 data for spin-up minus those using MERRA-2. Solid lines represent the model run started in January 2017 (S1) and dashed lines the one started in September 2017 (S2). Absolute differences (left y-axes) are shown for the cumulative differences of gross primary production (a), net primary production (b), the daily maximum differences of the canopy conductivity (c) and the green carbon pool (d). The right y-axes represent relative differences with respect to average annual sums of GPP (a) and NPP (b), and the overall maximum of modeled canopy conductivity (c) and the green carbon pool (d) in the years 2017 and 2018.

*occur for S1 within the first month, which is also the time of the largest soil water differences (Fig. 6). The MERRA-2 spin-up leads to a drier soil, which subsequently restricts stomatal opening and thereby limits the rate of photosynthesis and transpiration. Consequently, the reduced photosynthesis results in a smaller green carbon pool for the MERRA-2 spin-up (Fig. 1d). In the first two months after the start of the model run in January 2017, the differences in the green carbon pool amount to more than 80% of the annual average, with values remaining above 10% even after two years for S1. On the other hand, changes in the wood carbon pool occur over much longer time scales and may not reach equilibrium even after a 10-year spin-up and therefore the results should be interpreted with care. It is worth noting that the choice of spin-up data set causes differences in the order of 5% of the annual means, and these differences only slightly decrease throughout the two years of the model run (not shown). "*

The authors highlight and describe the key important results well. However, at some points, I am missing that these results should be discussed in greater detail and I am especially missing some I will outline these parts at the minor points.

**Minor points**

**3)** Lines 3-5: Introducing the abbreviations JSBACH, ERA5 and MERRA-2. Given the length of the abstract, I think it would also be nice to explain the abbreviations more. For example ERA5 reanalysis data? I would add one more sentence about what ERA and MERRA are.

A description of the names of the two reanalysis data sets was added to the abstract:
*"As a first step, we analyzed whether high-resolution global reanalysis data sets are suitable to be used as land surface model forcing. Namely, we used data from the fifth generation ECMWF atmospheric reanalysis of the global climate (ERA5) and the Modern-Era Retrospective analysis for Research and Applications, Version 2 (MERRA-2)."*

**4)** Lines 13: Great!

We think so, too!

**5)** Line 18: How would a separate canopy layer improve that? The way it is written that it appears to be quite speculative.

The sentence was rephrased to:
*"To tackle this issue, potential improvements can be made by improving the processes related to storage and vertical transport of energy. For instance, incorporating a distinct canopy layer into the model could be a viable solution."*

**6)** Line 32-35: I think this part should be at the end of the discussion.

The respective sentence is: "In this study, we use a site-level setup of the JSBACH LSM to evaluate the model performance at a rainforest site with a special focus on canopy processes." We assume that you propose to move this sentence towards the end of the introduction. We have chosen to present the overall goal of the study early on in the introduction to provide guidance to the reader throughout the subsequent paragraphs. Including this sentence enables a better comprehension of why we discuss previous research on topics like "forcing data sets for land surface models" and "model spin-up periods." Therefore, we prefer to keep this sentence at its original position within the text.

**7)** Line 44: Is JSBACH a land model or a land surface model? Choose one and use it consistently.

It is a "land surface model" and we now use this name everywhere in the text.

**8)** Line 74: there are two dots at the end of the sentence.

Done. See also 9).

**9)** Line 74-75: Something is missing in this sentence.

Something went wrong with the references to the sections. The sentences were corrected:
*"Results of the comparison between reanalysis data and ATTO measurements are presented in Sect. 3.1. Section 3.2 contains results of JSBACH model runs, which are used for sensitivity studies (Sect. 3.2.1 and 3.2.2) and to identify model shortcomings (Sect. 3.2.3), followed by a summary and conclusions in Sect. 4."*

**10)** Lines 75-76: There is no mention of the discussion in this paragraph.

The sentences were rephrased to:
*"Results of the comparison between reanalysis data and ATTO measurements are presented and discussed in Sect. 3.1. Section 3.2 contains results and a discussion of JSBACH model runs, which are used for sensitivity studies (Sect. 3.2.1 and 3.2.2) and to identify model shortcomings (Sect. 3.2.3), followed by a summary and conclusions in Sect. 5."*

**11)** Line 82: I would suggest using the word 'aspects' instead of 'features'.

Done.

**12)** Figure 1: While I like this figure I think it is needed to understand the key points of the manuscript and would suggest moving it to the appendix.

We believe that starting with the presentation of a map aids the reader in obtaining a comprehensive understanding of ATTO's surrounding area. Consequently, we prefer to keep this figure as part of the main text, rather than moving it to the appendix.

**13)** Table 1: I appreciate the effort of the authors putting together this list of measuring devices, but similar to Figure 1 I think it can be moved into the appendix.

The table was moved to the new "Appendix A: Meteorological instrumentation at the ATTO site".

**14)** Lines 102 (equation 1): These two equations are standard in atmospheric sciences and I would also suggest moving them and the explanation to the appendix.

We now write *"We convert the measured relative humidity to specific humidity and the air pressure measured at 81 m height to surface pressure. More details are presented in Appendix B."*; and moved the equations and the description to the new "Appendix B: Humidity conversion".

**15)** Line 122 Section 2.2: Why did the authors choose ERA5 and MERRA-2? Why not any of the other datasets? There should be 1-2 sentences about why such datasets are preferred over other available ones.

We added the following sentences:
*"We utilize only those global reanalysis data sets that provide data at least hourly, which enables us to analyze diurnal cycles. Specifically, the selected data sets include ERA5 (Sect. 2.2.1) and MERRA-2 (Sect. 2.2.2), which also have a relatively high spatial resolution of less than 70km."*

**16)** Lines 165-169: This part is rather a model setup than a model description and I suggest moving it to the end of this subsection.

Done.

**17)** Line 174: What is a T63 grid?

We use a JSBACH version that is part of the MPI-ESM model (Mauritsen et al., 2019), which uses a spectral grid. T63 means a truncation of the grid to 63 wave numbers, which corresponds to 192 x 92 grid points. We added this information to the manuscript:
*"These characteristics are based on the data set of land surface parameters derived by Hagemann (2002) on a T63 spectral grid with 192x98 (lon,lat) grid points, which corresponds to a grid cell size of about 200 km at the considered latitude."*

**18)** Line 199: What does 'almost perfectly agree' mean? Such statements should also be supported by some statistics like RMSE.

Both annual means from ERA5 and ATTO are 26.1 °C. RMSE values were added for both ERA5 and MERRA-2:
*"ERA5 annual mean temperatures at 10 m height between 2014 and 2018 agree almost perfectly with the ATTO values measured at 18 m above the forest - both with mean values of 26.1°C and an RMSD of 1.4 K. Compared with ATTO measurements, MERRA-2 is generally too warm with annual average temperatures of 26.9°C and a larger RMSD of 2.0 K."*

**19)** Line 206: See major points. Why start by explaining Fig. 3c?

The figures have been rearranged. Please see 1) for details.

**20)** Line 230: The authors describe the bias in windspeed that they find. I am missing the implications for LSMs if the windspeed is much lower or higher. What should that do in theory to LH and SH and canopy temperatures? Or is that hard to say at all?

We added the following explanation:
*"Since turbulent heat fluxes scale with wind speed, an underestimation of the latter would initially result in an decrease of sensible and latent heat fluxes, which then increases the surface temperature. However, a higher surface temperature increases sensible heat fluxes and thus the overall impact on surface and soil temperatures is difficult to estimate."*

**21)** Line 235: This statement about generality needs a reference.

A similar statement can be found in the next paragraph concerning diurnal cycles where we state that: *"... ERA5 overestimates the maximum rain rate by about 40 %. ERA5 also shows a negative shortwave radiation bias during the day (Fig. 4e), which indicates an overestimation of the cloud cover."* To avoid repetition, we thus simply deleted the sentence you referred to.

**22)** Line 252: Can you test if that is true or not?

The regional patterns of early morning precipitation were shown in Fig. A1 based on IMERG data. We now added the corresponding maps also based on ERA5 and MERRA-2 data (see Fig. 2). A paragraph describing the new figure was added to the appendix:
*"While there are notable differences between LN-EM and EM precipitation (Fig. A1a and d) for IMERG, the patterns for the two reanalysis data sets stay roughly the same before and after sunrise. Fig. 4d indicates that MERRA-2 does not capture the second EM maximum of*

[Figure]

Figure 2: Fraction of nighttime precipitation between (top) 0:00 and 8:00 and (bottom) 6:00 and 8:00 local time averaged over the wet season (JFMA) from 2014 to 2018 based on the IMERG precipitation data set (a,d), MERRA-2 (b,e) and ERA5 (c,f) data. The circle marks the ATTO location. Red and orange boxes denote the considered grid boxes for ERA5 and MERRA-2, respectively (see also Fig. 1).

*precipitation, which is observed in the ATTO measurements. It is also evident from Fig. 4c and f, that ATTO is located at an area with relatively low fractions of nighttime precipitation. Fractions are much larger at about 100 km further to the east. There, the maximum of the diurnal cycle of precipitation is between 7:00 and 10:00 local time (not shown), which agrees better with the observed ATTO results. This gives a hint that MERRA-2 does not generally fail to reproduce early morning precipitation, but partly produces it at the wrong location."*
A short explanation was also added to the respective paragraph in Sect. 3.1.2:
*"To further analyze the occurrence of the early morning peak we evaluated rainfall data from the IMERG data set in a larger region around ATTO (for details see Appendix A) and compared the results to data from the two reanalyses. ... For MERRA-2, the analysis of regional patterns of early morning precipitation reveals that a morning maximum can be found at grid points located about 100 km to the east. This gives a hint that MERRA-2 might just produce early morning precipitation at a slightly wrong location. "*

**23)** Figure 5: What kind of measurement uncertainty is used? How can it be so small given we are looking at rainrates over 5 years?

The measurement uncertainty of the rain gauge given by the manufacturer is 2 to 3 % (see Tab. 1) and this value is also used to calculate the shaded area in Fig. 5. If the errors of each hourly measurement were randomly distributed, averaging over several years would even decrease the uncertainty range. By assuming the worst case, meaning that measurements always overestimate (underestimate) the true precipitation rate by 3 %, we obtain the upper (lower) border of the shaded area.

**24)** Line 258: I think the specific humidity (Fig. 3b) deserves more attention. Why are both models going down while the observations are going up in August-October?

This is now described in more detail:
*"It is most striking that the diurnal cycles show distinctly different shapes, with a maximum of the humidity measured at ATTO in the later morning in the dry season and later in the afternoon in the wet season. In contrast, specific humidity values of both reanalyses start to decrease in the morning with a (local) minimum in the early afternoon. For ERA5, this general underestimation of daytime specific humidity is the reason for the underestimation of monthly means in all months observed in Fig. 5d. For the same reason, MERRA-2 underestimates monthly mean humidity in the dry season. However, the diurnal cycle in the wet season indicates that MERRA-2 humidity is always about $0.9\,g\,kg^{-1}$ larger than ERA5 humidity. Thus, the overestimation of nighttime humidity values compared to ATTO measurements compensates the underestimation in the afternoon, resulting in a negligible overall bias of monthly means in the wet season. "*

**25)** Line 270: ATTO does not have that radiosonde instrument, right? I would add why we can't do that in that study.

Yes, during the considered time period there were no regular radiosonde measurements. We added this information to the text:
*"Testing this hypothesis, however, would require more investigations with measurements spanning the whole ABL column, e.g. from radiosonde measurements. Such measurements are unfortunately not a part of regular measurements conducted at the ATTO site."*

**26)** Line 274-277: This describes what is happening in the models used for reanalysis data, right? Why does this not apply to the 'reality' or the ATTO tower?

As we state in the paragraph following these lines, there are two possible mechanism, which could be responsible for the different shape of the humidity diurnal cycle observed at ATTO: 1) evapotranspiration is stronger than modeled by the two reanalyses, or 2) vertical mixing-processes within the ABL are weaker than in the reanalyses. We rephrased the respective paragraphs for clarity:

*"The diurnal cycles of specific humidity showed a maximum during midday for ATTO measurements but a minimum for both reanalyses. The processes leading to such a humidity minimum during midday have been described ... This process is modeled by the reanalyses, but it appears that this is not what happens in reality at the ATTO site. The different observed shape of the diurnal cycle with a maximum at midday could have two possible reasons: 1) evapotranspiration is stronger than modeled by the two reanalyses, or 2) vertical mixing-processes within the ABL are weaker than in the reanalyses."*

**27)** Line 293: Typo: biases instead of biased

Done.

**28)** Line 298: How has the data been optimized? Should be described in the method section.

Due to the dependence of the optimized forcing on the outcomes of Sections 3.2.1 and 3.2.2, it would be challenging to comprehend if we were to describe it solely in the methods section. The details about the optimized forcing are described in the first paragraph of Sect. 3.2.3. We now include a reference to this paragraph in the respective sentence:

*"Based on the conclusions of these two sections we then set up a model run with optimized forcing, which is based on bias-corrected ERA5 data and ATTO measurements (see Sect. 3.2.3 for details)."*

**29)** Line 300: Section 3.2.1: Are there any other impacts on e.g. GPP or NPP or transpiration or stomatal conductance in general?

We added results for GPP, NPP, canopy conductance, as well as woody and green carbon pools. See 2) for details.

**30)** Line 310: Why is that expected?

The sentence was rephrased to:
*"The largest differences occur in the deeper soil layers, which is expected considering the longer adaptation time required for deeper soil layers to respond to changes in surface forcing."*

**31)** Figure 7: The description of this figure is confusing. I recommend adding ERA5 and MERRA to the legends. I also recommend explaining each subfigure in alphabetical order.

Generally, the figure shows model result using ERA5 as spin-up forcing minus those using MERRA-2. Thus, it would not be helpful to add "ERA5 - MERRA-2" to every single figure legend. Instead, we thoroughly rephrased the caption for clarity:
*"Consequences of different spin-up data sets on soil conditions: model results using ERA5 data for spin-up minus those using MERRA-2. Time series of differences for (a) soil temperature, (b) soil water content at different depths and (c) RMSD for turbulent fluxes. Corresponding differences between ERA5 and MERRA-2 spin-up relative to model biases (differences between model results and measured ATTO data, see also Sect. 3.2.3) are shown in panels (d) for soil temperature and (e) for soil water. Results are presented for two different starting times of the model run after spin-up in the wet (S1: January 2017) and dry (S2: September 2017) seasons. Empty symbols in (d) and (e) indicate that differences between ERA5 and MERRA-2 spin-up are below 0.2 K for soil temperature or below 0.7 % for soil water content (for S1 compare values in (a) and (b), respectively)"*
In addition, the figure panels were renamed and partly rearranged to ensure that they now

appear in alphabetical order in the caption.

**32)** Figure 9a) Missing legend for JSB.

JSB was added to the legend.

**33)** Line 443: Figure 13a,b does not support that statement before!

The green bars in Fig. 13a,b clearly indicate that the data availability is about 80 % during the night and drops to about 30 to 40 % during midday and thus supports the statement. For clarity, we added these numbers to the text. We also deleted the reference to Fig. 13 to avoid mentioning Fig. 13 before Fig. 12 in the text.
*"It needs to be kept in mind that data availability is much lower during the day (30 to 40 % at midday) than at night (about 80 %) and thus the diurnal cycles should be interpreted with care."*

**34)** Line 495: I like the summary of the results. However, here I am missing the implications for the vegetation. Why should we care about these biases? If there are almost no roots in the deepest soil layers, is a 20% difference important? Again: are there no other biases for other variables?

As we stated in the paper, the root depth at the considered grid point is 1,95 m, which means that the plants actually do have access to the deepest soil layer between 1.23 and 2.23 m. To elaborate further on the implications for the vegetation, we added the following statement regarding the biases of other variables:
*"The differences for deeper soil temperatures and the green carbon pool remained non-negligible even after one year. For other variables associated with plant growth (GPP, NPP, canopy conductance), noticeable differences were observed for up to six months after the start of the model run."*

**References**

Mauritsen, T., Bader, J., Becker, T., Behrens, J., Bittner, M., Brokopf, R., Brovkin, V., Claussen, M., Crueger, T., Esch, M., Fast, I., Fiedler, S., Fläschner, D., Gayler, V., Giorgetta, M., Goll, D. S., Haak, H., Hagemann, S., Hedemann, C., Hohenegger, C., Ilyina, T., Jahns, T., Jimenéz-de-la Cuesta, D., Jungclaus, J., Kleinen, T., Kloster, S., Kracher, D., Kinne, S., Kleberg, D., Lasslop, G., Kornblueh, L., Marotzke, J., Matei, D., Meraner, K., Mikolajewicz, U., Modali, K., Möbis, B., Müller, W. A., Nabel, J. E. M. S., Nam, C. C. W., Notz, D., Nyawira, S.-S., Paulsen, H., Peters, K., Pincus, R., Pohlmann, H., Pongratz, J., Popp, M., Raddatz, T. J., Rast, S., Redler, R., Reick, C. H., Rohrschneider, T., Schemann, V., Schmidt, H., Schnur, R., Schulzweida, U., Six, K. D., Stein, L., Stemmler, I., Stevens, B., von Storch, J.-S., Tian, F., Voigt, A., Vrese, P., Wieners, K.-H., Wilkenskjeld, S., Winkler, A., and Roeckner, E.: Developments in the MPI-M Earth System Model version 1.2 (MPI-ESM1.2) and Its Response to Increasing CO2, Journal of Advances in Modeling Earth Systems, 11, 998–1038, https://doi.org/https://doi.org/10.1029/2018MS001400, 2019.

---

## Author Comment (AC2)

We thank the anonymous reviewer for the time and effort they invested in critically reviewing our manuscript. Please find answers to your very helpful comments and suggestions below.

**Answers to reviewer 2**

**Major points**

**1)** Sect. 2.3.2: Do I understand it correctly that JSBACH is run in offline mode at one (grid) point, using an atmospheric forcing from observation or reanalyses? Then, why is a spin-up of ten years needed? For instance, Chen et al. (1997) have shown that in PILPS Phase 2a the vertical profiles of soil temperature and water content converged to an eqilibrium after two to three years. This was in Cabauw (The Netherlands), not in the rainforest. Anyway, since there is an extensive rain period, you could start the simulation there, initialize the soil water content at saturation, and run the model into equlibrium. You should show that this is not already reached after two or three years, but that you really need ten years. This would make more sense in order to understand the model behaviour of JSBACH. As you say, the soil is not even particularly deep. Maybe, with a shorter spin-up, you find periods of the ATTO measurements with less data gaps?

We did not mean to suggest that all future studies of this type should use a 10-year spin-up period. In fact, our results in Sect. 3.2.1 indicate that for many of the considered variables, a spin-up period of less than 1 year would be sufficient. After longer time periods, we would only expect additional changes for soil temperatures in the deeper layers and for carbon pools (see new results in point 2 of reviewer 1) at this specific site.

While it would have been possible to choose a time period with two or three years of nearly complete ATTO measurements for the spin-up forcing, we made the decision to incorporate reanalysis data as well. This approach allows for greater versatility, as it can be utilized in other studies where only data from shorter campaigns, spanning a few weeks or months, are accessible. By incorporating reanalysis data into the spin-up, our setup becomes applicable in a wider range of scenarios. We now added a discussion of these points in the first paragraph of Sect. 3.2.3:

*"Based on the results of the previous sections we construct an optimized version of the spin-up run. The findings from Sect. 3.2.1 indicate that the duration during which the choice of spin-up forcing data set has a significant impact on most variables is less than one year. As a result, a spin-up period of two or three years would be sufficient to reach an equilibrium state for soil water content and soil temperatures in the upper layers at this specific site. However, variables like temperatures of the deeper soil layers or the green carbon pool require a longer spin-up duration. Therefore, when employing a standalone land surface model, the selection of the spin-up period should be determined by the specific processes of interest. In our case, we adopt a cautious approach and use a 10-year spin-up period for the model, which has the following characteristics: ... "*

**2)** Sect. 3.3.1: The wind looks difficult. Likely, there is a problem with the representativity, since the reanalyses can not describe the high forest canopy, but in reality it exists. It is not clear to me how a bias correction should work here. Maybe, it is mainly a matter of finding the correct heights above ground (or canopy top) to make the quantities comparable? Could you discuss this a bit more?

We assume that you are referring to Sect. 3.2.2 "Sensitivity to wind speed and precipitation biases". While it is evident that reanalyses lack a distinct canopy layer and therefore cannot fully capture all relevant processes, the forest canopy's influence on the wind profile is partially considered by increasing the roughness length. However, considering the biases identified in other studies in regions with different terrains and land uses (as listed in Sect. 3.1.1), it is

more plausible that the bias is related to a general issue with the wind representation in the reanalyses, rather than simply a discrepancy in height alignment of the wind profiles.
Nonetheless, we acknowledge that the approach we used in Sect. 3.2.2, where we applied an offset to correct the biases in wind speed, is an oversimplification. The main objective of this section was to demonstrate the potential impact that biases of this magnitude could have on the model results. We added a short description of this limitation to the first paragraph of Sect. 3.2.2:

*"The results presented in Sect. 3.1.1 indicate that the underestimation of the wind speed by the two reanalyses is a complex issue. For simplicity, we apply a very simple bias correction in this sensitivity study by adding an offset of the annual mean wind speed bias of -1.2 $ms^{-1}$ between 2014 and 2018 to the MERRA-2 data. The results are then compared to those using the original MERRA-2 forcing."*

**3)** L. 472-474: Instead of "temperature damping effect" I would call it rather "shading effect". Without a vegetation canopy the model has no chance to get any of the following temperatures right: Soil temperature (different depths), surface temperature, and 2-m temperature. Instead of a complex canopy scheme, a simpler way to represent this mechanism is e.g. a conceptional "skin temperature" scheme, see e.g. Viterbo and Beljaars (1995), Heidkamp et al. (2018), or Schulz and Vogel (2020). The work of Heidkamp et al. (2018) is available in JSBACH. It may be advisable to apply it in your study, in order to represent the shading effect due to the vegetation. This would reduce the amplitudes of the diurnal cycles of the soil temperatures, and increase the amplitude of the surface and 2-m temperature. It would be good if you could demonstrate this in your manuscript, because the observations you have available.

We selected the term "temperature damping effect" to provide a straightforward description of the fact that the diurnal cycle of the surface temperature actually looks like soil temperature from a deeper layer, where the amplitude is dampened. While the term "shading effect" you proposed is indeed applicable in this context, it may primarily invoke associations with radiation-related processes and might not encompass the crucial aspect of canopy heat storage, which is also important for this effect. Consequently, we believe it is more appropriate to adhere to the original term of "damping".
In general, we agree with your remark concerning the "skin temperature" scheme. A simple approach, which includes only the canopy heat storage instead of an explicit representation of the canopy layer, would likely be able to capture at least a part of the dampening/shading effect. Unfortunately, the approach by Heidkamp et al. (2018) was originally implemented in JSBACH 3 (version 3.11) and has not yet been transferred to version 4 of JSBACH, which we used for the model runs in this study. Nevertheless, we have expanded discussion of the dampening effect and write now:

*"However, JSBACH (version 4) does not include an explicit canopy layer or a parametrization of the canopy heat storage effect. Consequently, the model is not able to capture this dampening effect." (original l. 438)*
*"Secondly, it would be beneficial to include a representation of the canopy heat storage effect into the model. This could be accomplished by modeling the processes in a separate canopy layer explicitly or by adopting a simpler approach that parametrizes the heat storage by the canopy. Heidkamp et al. (2018) and Schulz and Vogel (2020) demonstrated that a simple approach, which is based on a skin temperature formulation, reduces the underestimation of the amplitude of the diurnal cycle of surface and soil temperatures and the corresponding incorrect phase shifts. Moreover, the skin temperature formulation improves biases in latent and sensible heat fluxes (Schulz and Vogel, 2020; Renner et al., 2021)." (original ll. 472)*

**4)** L. 475-476: The evaporation of water from the interception reservoir is usually less relevant for the simulated soil temperature. It may play a role after dew fall in the morning (or after

rain fall) for the 2-m temperature. Please rephrase.

The sentences were rephrased to:
*"To reduce the soil temperature bias of the model, it might also be beneficial to re-evaluate the representation of additional cooling terms within the canopy layer. For example, evaporation of dew or of rainfall intercepted by the vegetation impacts near-surface air temperatures, which then in turn influence soil temperatures."*

**Minor points**

**5)** L. 49: ... errors in the ...

Done.

**6)** L. 54: ... based on two ...

Done.

**7)** L. 58: ... (Yang et al. 1995), ...

Done.

**8)** L. 66: ... of the forcing data ...

Done.

**9)** L. 71: ... turbulent heat fluxes ...

Done.

**10)** L. 72: shortcomings

Done.

**11)** L. 74: Section numbers are missing

Done.

**12)** L. 90: less or equal, or larger or equal 36 m?

It was corrected to $\geq$36m.

**13)** Fig. 9: (a) and (b) are mixed

The panels were re-aranged to match the order of appearance in the text and the figure caption was corrected.

**14)** L. 398: "field" capacity. Anyway, field capacity is not saturation, this would be pore volume. Please rephrase.

The sentences were rephrased to:
*"In the wet season, all soil layers approach a constant value of about 35 %, which is close to the field capacity of the soil. This indicates that the soil water content approaches saturation levels during rainfall events."*

**15)** L. 404: soil types $\rightarrow$ soil textures

Done.

**16)** L. 407: an exponential root profile would be typical

We added the following sentence:
*"Other studies indicated that it is beneficial to adopt an exponential root profile assumption (e.g. Jackson et al., 1996; Zeng, 2001)"*

**17)** L. 468 and around: soil types → soil textures

Done.

**References**

Heidkamp, M., Chlond, A., and Ament, F.: Closing the energy balance using a canopy heat capacity and storage concept–A physically based approach for the land component JSBACHv3. 11, Geoscientific Model Development, 11, 3465–3479, https://doi.org/10.5194/gmd-11-3465-2018, 2018.

Jackson, R. B., Canadell, J., Ehleringer, J. R., Mooney, H., Sala, O., and Schulze, E.-D.: A global analysis of root distributions for terrestrial biomes, Oecologia, 108, 389–411, https://doi.org/10.1007/BF00333714, 1996.

Renner, M., Kleidon, A., Clark, M., Nijssen, B., Heidkamp, M., Best, M., and Abramowitz, G.: How well can land-surface models represent the diurnal cycle of turbulent heat fluxes?, Journal of Hydrometeorology, 22, 77–94, https://doi.org/10.1175/JHM-D-20-0034.1, 2021.

Schulz, J.-P. and Vogel, G.: Improving the processes in the land surface scheme TERRA: Bare soil evaporation and skin temperature, Atmosphere, 11, 513, https://doi.org/10.3390/atmos11050513, 2020.

Zeng, X.: Global vegetation root distribution for land modeling, Journal of Hydrometeorology, 2, 525–530, https://doi.org/10.1175/1525-7541(2001)002¡0525:GVRDFL¿2.0.CO;2, 2001.